# InfoAgent: Advancing Autonomous Information-Seeking Agents

## ABSTRACT

Building Large Language Model agents that expand their capabilities by interacting with external tools represents a new frontier in AI research and applications. In this paper, we introduce InfoAgent, a deep research agent powered by an innovative data synthesis pipeline and orchestrated web search tools. To construct challenging, hard-to-find queries, we build entity trees and apply sub-tree sampling with entity fuzzification to systematically increase question difficulty. Unlike prior work that relies heavily on commercial search tools, we develop a dedicated self-hosted search infrastructure, enhancing transparency of agent environments and facilitating further advancement of agent capacity. We evaluate the effectiveness of our data pipeline by measuring the average number of tool calls required to correctly answer a question, and also show that our agent yields better performance when equipped with our tools. Our InfoAgent is post-trained from Qwen3-14B using a two-stage recipe: cold-start supervised finetuning to instill long-horizon search behaviors, followed by reinforcement learning which significantly improves reasoning-driven tool use. With our methods, InfoAgent achieves 15.3% accuracy on BrowseComp, 29.2% on BrowseComp-ZH, and 40.4% on Xbench-DS, outperforming prior open-source deep research agents such as WebSailor-72B and DeepDive-32B.

## 1 INTRODUCTION

The Internet has revolutionized the way people acquire knowledge, yet the tools that mediate access to online information have evolved unevenly (Zhang et al., 2025). Recently, researchers have enhanced Large Language Models (LLMs) with agentic capabilities via Reinforcement Learning (RL), which allows them to autonomously plan, search, and learn in an ongoing loop (OpenAI, 2025b). Deep Research Agents (DRAs) are distinguished by their ability to plan, reason, execute multi-step information-seeking actions, such as retrieving documents from the Internet via given tools, and complete complex research tasks. Recognizing their potential, major AI providers have raced to deliver commercial implementations (OpenAI, 2025a; Perplexity, 2025; xAI, 2025a; Google, 2025). This phenomenon shows that deep research is becoming a defining feature of next-generation information platforms.

The implementation of DRA faces two challenges: effective strategy for data synthesis and the establishment of an efficient interactive environment. Existing open-source DRAs often perform shallow searches, mainly because they are trained on relatively simple data (Jin et al., 2025; Li et al., 2025c). Training dataset must encompass a broad range of data, which is of various uncertain types, so that the agent is forced to link disparate pieces of information and infer new knowledge when retrieving documents. Meanwhile, some agents are trained in simulated environments, which are underpowered when confronted with challenging real-world problems (Jin et al., 2025). We suggest that RL infrastructure for DRA must provide tools accessible to real-world information, which must be able to handle high-concurrency search and browsing calls, and returns consistent results during RL training.

In this paper, we build InfoAgent, a DRA designed for long-horizon information seeking and deep reasoning. Our work addresses the two bottlenecks identified above, data synthesis and interactive environment. On the data side, we devise an end-to-end synthesis pipeline that generates challenging problems that must be solved via deep research. Considering the requirements for broader logical

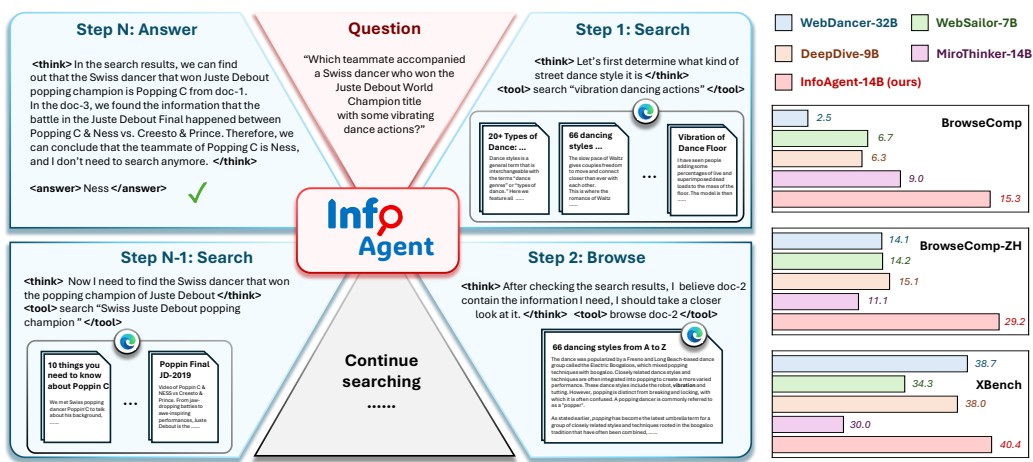

Figure 1: An illustration of how InfoAgent leverage search and browse tools to solve information-seeking problems (left). InfoAgent achieves advanced results on several deep research benchmarks among small open-source agents (right).

structures and more varied uncertain types in training data, we start from raw Wikipedia entity set and build entity trees. Then, we apply sub-tree sampling with entity fuzzification to systematically enhance the difficulty of each question. These designs force an agent to perform long-horizon retrieval and conjunctive reasoning rather than rely on single-hop lookup, thereby lifting the upper bound of its ability. Regarding the environment, many existing agents lean heavily on commercial search APIs. In this case, the retrieval process is hidden behind proprietary services, and the efficiency is also constrained by external rate limits and tool availability. This dependence not only makes the behavior of the agent uncontrollable, but also makes training and evaluation hard to reproduce. We therefore forgo commercial services and construct our own search and browsing infrastructure. This gives us fine-grained control over outputs and provides a transparent experimental environment.

We evaluate our data and environment by post-training the Qwen3-14B (Yang et al., 2025) with a two-stage recipe. In the first stage, we perform supervised finetuning (SFT) as a cold start, in order to instill long-horizon search behavior into the model. In the second stage, we apply RL to refine its ability of reasoning-driven tool use. As shown in Figure 1, InfoAgent attains 15.3% accuracy on BrowseComp (Wei et al., 2025), outperforming previous open-source DRA such as WebSailor-72B (Li et al., 2025b) and DeepDive-32B (Lu et al., 2025). Moreover, even though all of our synthetic training data are in English, InfoAgent exhibits strong cross-lingual generalization: it achieves competitive accuracy on the Chinese benchmarks, including an accuracy of 29.2% on BrowseComp-ZH (Zhou et al., 2025b) and 40.4% Xbench-DeepResearch (Chen et al., 2025a). Our model not only achieves first place on multiple benchmarks (e.g., BrowseComp, BrowseComp-ZH, WebWalkerQA) among models with fewer than 15B parameters, but also surpasses some 32B and 72B models. Thanks to our high-quality synthesized data and efficient, stable training environment, InfoAgent demonstrates strong performance even when driven by a small 14B parameter model. We will release our open-source data and environment to further promote the development of this field.

## 2 RELATED WORKS

**Reasoning Model.** The emergence of ChatGPT (OpenAI, 2022) has brought the general reasoning capabilities of LLMs to widespread public attention. In the period that followed, several improved foundational model series such as Grok, Gemini, Claude, and GPT were proposed. These models have validated the scaling laws (Kaplan et al., 2020) and consistently set new records on reasoning benchmarks. As the marginal effects of pre-training diminish, works such as o1 (Jaech et al., 2024) and Deepseek-R1 (Guo et al., 2025) have found that post-training with reinforcement learning to increase the reasoning length during test-time further enhances the model's ability to tackle exceptionally difficult problems. Today, advanced reasoning models like Gemini-2.5-pro (Comanici et al.,

2025), o3 (OpenAI, 2025c), Claude-4 (Antropic, 2025), and Grok-4 (xAI, 2025b) have achieved gold-level performance in top-tier human mathematics (AoPS, 2025) and coding (Jimenez et al., 2023; Miserendino et al., 2025) competitions. Such reasoning capabilities should theoretically drive evolutionary changes in societal productivity. We believe that the gap lies in the model's reasoning abilities and its interaction level with reality. Models need to use tools to interact with the real world to maximize the practical impact of their powerful reasoning capabilities.

**Retrieval-augmented Generation (RAG).** RAG refers to models that combine a pre-trained language generator with an explicit retrieval component (Lewis et al., 2020; Borgeaud et al., 2022; Guu et al., 2020). Typically, RAG systems segment data and employ some form of vector retrieval for information access (Borgeaud et al., 2022; Chen et al., 2022; Gong et al., 2020; Ishiwatari et al., 2017). RAG often integrates the retriever as a component within the system, allowing for end-to-end training alongside the generator (Guu et al., 2020; Izacard et al., 2023; Lewis et al., 2020). Compared to information-seeking agents, most RAG systems demonstrate high efficiency because RAG either treats the retrieved documents as latent variables (Li et al., 2021; Wang et al., 2023; Févry et al., 2020) or directly injects the documents into prompts. In summary, RAG retrieval is efficient, static, and localized, whereas the agent studied in this paper is iterative, offering greater flexibility. This allows it to handle data from multiple sources from a wider range of tools.

**Deep Research Agents and Benchmarks.** Recent researches bring web-browsing into agentic model, such as DeepDive (Lu et al., 2025), WebSailor (Li et al., 2025b), WebSailor-V2 (Li et al., 2025a), and ASearcher (Gao et al., 2025). They leverage data-synthesis methods like InfoSeek (Xia et al., 2025), which enhance long-horizon browsing via knowledge-graph question synthesis, scalable multi-turn RL, and large-scale data generation. Search-augmented frameworks including Search-o1 (Li et al., 2025c), Search-R1 (Jin et al., 2025), DeepResearcher (Zheng et al., 2025), and the multi-agent WebThinker (Li et al., 2025d) integrate *think–search–write* loops in real-web environments. Memory-based continual agents such as AgentFly (Zhou et al., 2025a) achieve advanced performance without tuning base models. At the foundation-model level, GLM-4.5 (Zeng et al., 2025a) and Kimi-Researcher (MoonshotAI, 2025) adopt multi-stage post-training with RL to strengthen tool use and reasoning. Benchmarks for DRAs include BrowseComp (Wei et al., 2025), HLE (Phan et al., 2025), BrowseComp-ZH (Zhou et al., 2025b) xBench (Chen et al., 2025a), GAIA (Mialon et al., 2023), and Mind2Web 2 (Gou et al., 2025). They provide more complex assessment that require integration of more sources of information compared to traditional RAG and knowledge-based QA benchmarks.

## 3 INFOAGENT APPROACH

We adopt the ReAct framework (Yao et al., 2023) to construct InfoAgent, which iteratively combines reasoning with tool calls to arrive at a solution. For a given problem, the agent engages in an action cycle: at each step, it incorporates new observations from the tool, generates reasoning traces, and continually calls tools with the corresponding arguments. Both the reasoning process and tool outputs are appended to the context of the LLM, enabling continuous research. For our InfoAgent, we provide two web-based tools, search and browse, to process information-seeking requests. The search tool retrieves a ranked list of web URLs along with content snippets, while the browse tool enables deeper investigation of the content associated with a given URL.

Below, we introduce our method to synthesize data and the implementation of tools, which enable effective and efficient training for InfoAgent.

### 3.1 DATA SYNTHESIS PIPELINE

We introduce a two-stage pipeline to automatically synthesize complex, multi-entity search questions, which can be solved only if the model excels in long-horizon search and reasoning. This pipeline converts raw Wikipedia entities into structured QA pairs via (1) *Tree Construction* and (2) *QA Generation* shown in Figure 2.

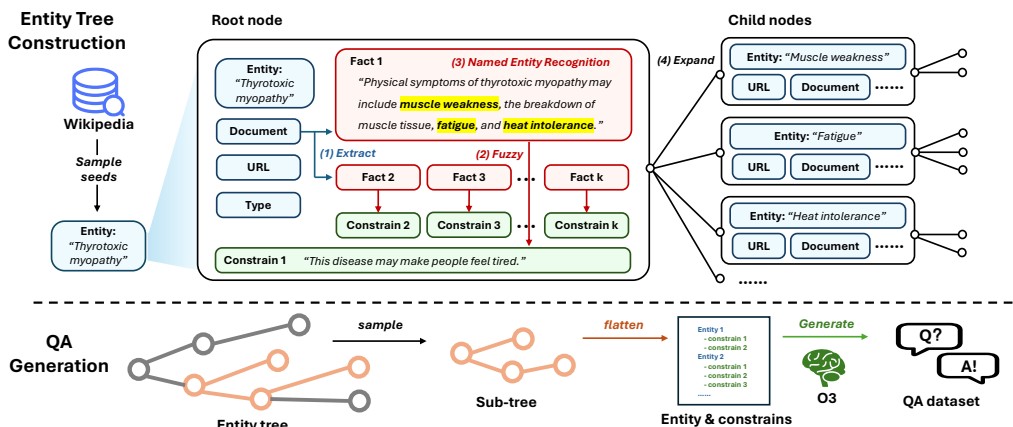

Figure 2: Pipeline for synthesizing multi-entity search questions. (1) *Tree Construction* converts Wikipedia entities into fuzzy-annotated tree structures with configurable branching. (2) *QA Generation* samples sub-trees and produces questions.

**Tree Construction.** Given a filtered Seed Set $\mathcal{S} = \{\texttt{entity}_i = (\texttt{name}_i, \texttt{url}_i)\}_{i=1}^N$ from Wikipedia, each entity is transformed into a node $v_i$:

$$v_i = (\texttt{name}_i, \texttt{url}_i, \mathcal{F}_i), \quad \mathcal{F}_i = \{f_k\}_{k=1}^m \tag{1}$$

where $\mathcal{F}_i$ is a set of facts extracted from the corresponding $\texttt{url}_i$. To construct the entity tree $T_i$, child entities $\mathbf{C_i}$ can be recognized from each fact by $\bigcup_{k=1}^m \mathbf{NER}(f_k)$, where $\mathbf{NER}$ is Named Entity Recognition. We randomly select a certain number of entities from $\mathbf{C_i}$ to build child nodes for $v_i$, determined by tree structure configuration $\Theta$. Once the child nodes for $v_i$ are created, they will be recursively expanded to an entity tree $T_i$. Finally, we get an entity forest $\mathbb{T} = \{T_i\}_{i=1}^N$.

To encourage multi-evidence reasoning, facts in each node are fuzzified through three stages:

$$\widetilde{\mathcal{F}} = \underbrace{f_{\text{llm}}}_{\text{semantic rephrasing}} \circ \underbrace{f_{\text{static}}}_{\text{numbers/dates} \rightarrow \text{ranges}} \circ \underbrace{f_{\text{entity}}}_{\text{entity fuzzy}} (\mathcal{F}). \tag{2}$$

First, $f_{\text{entity}}$ substitutes specific entity names with generic descriptions (e.g., "Albert Einstein" becomes "a famous physicist"). Then, $f_{\text{static}}$ replaces specific numbers and dates with broader ranges or vague descriptions (e.g., "1992" becomes "early 1990s", "42" becomes "around 40"). Finally, $f_{\text{llm}}$ rephrases the facts by LLM, which modifies the expression but keeps its meaning unchanged. This process guarantees that entity names and specific dates/numbers are replaced with their fuzzy counterparts, making it harder to identify the entity based on internal knowledge or direct search.

After fuzzification, for each node $v$, we construct a constraint set $\mathcal{K}(v)$ from fuzzified facts $\widetilde{\mathcal{F}}(v)$ by solving the following optimization problem:

$$\min_{\mathcal{K} \subseteq \widetilde{\mathcal{F}}(v)} \left| \mathbf{q}(\mathcal{K}) \right| \quad \text{s.t.} \quad v \notin \mathbf{p}(\mathcal{K}). \tag{3}$$

Here, $\mathbf{q}(\mathcal{K})$ denotes the estimated *ground-truth entity space* consistent with all constraints in $\mathcal{K}$, while $\mathbf{p}(\mathcal{K})$ represents the *candidate entity space* retrieved by a powerful agent using only shallow cues from $\mathcal{K}$. Intuitively, $\mathcal{K}(v)$ is the resultant constraint set that is *barely sufficient* to solve the problem after filtering and screening out easy constraints. Our objective $\min_{\mathcal{K}} |\mathbf{q}(\mathcal{K})|$ subject to $v \notin \mathbf{p}(\mathcal{K})$ captures the core intuition: we search for constraint sets where few entities satisfy all constraints, yet a shallow-cue retriever still fails to identify the true entity $v$. This separation prevents shallow pattern matching and naturally encourages sustained, evidence-based search.

In practice, we implement this idea through a controlled sampling procedure. After each sampling attempt, we prompt a strong agent (o3 with web tools) to identify the correct entity given only the current constraint set $\mathcal{K}$, and we apply an answer verifier to check whether it can recover the true entity name. If o3 succeeds, we further ask it to select a minimal subset of constraints $\mathcal{K}_c$ that is sufficient to solve the task, which implies $v \in \mathbf{p}(\mathcal{K}_c)$. We then remove these overly informative

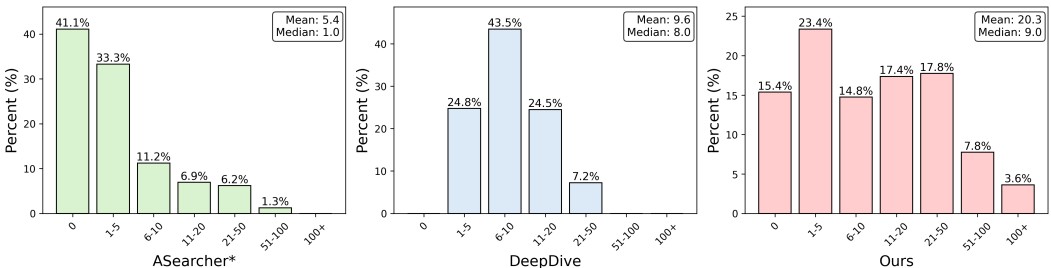

Figure 3: Tool call distribution analysis on randomly sampled subsets ($n$=800 per dataset). Our dataset exhibits both more tool calls and broader distribution compared to ASearcher and DeepDive.

constraints from the pool and resample a new candidate set from the remaining constraints. If o3 fails, or if the number of remaining constraints falls below a predefined threshold, we accept the sampled $\mathcal{K}$ as $\mathcal{K}(v)$.

**QA Generation.** From the forest $\mathbb{T}$, we sample multiple sub-trees. For each sub-tree with root node $v_{root}$, we generate a question $q_u$ about an attribute $\phi(v_{root})$ of $v_{root}$ rather than $v_{root}$ itself. This strategy increases the difficulty of the question and verifiability of the answer.

$$q_u = \mathcal{G}(\{\mathcal{K}(v)|v \in T\}, answer), \qquad answer = \phi(v_{root}), \qquad (4)$$

where $\mathcal{G}$ is the question generator based on LLM. The question and corresponding answer pair form the question set $\mathcal{Q}_0$. Due to the fuzzification, the generated questions may not have unique correct answers. Following BrowseComp (Wei et al., 2025), we employ o3 to conduct multi-round testing on $\mathcal{Q}_0$, filtering out questions that o3 fails to solve in all attempts. This process yields a refined question set $\mathcal{Q}$ with high difficulty and guaranteed resolvability.

**Synthesis of Reasoning Trajectories.** We generate high-quality trajectories that correctly solve the problems in $\mathcal{Q}$, using the advanced model o3 (OpenAI, 2025c) in a ReAct-style (Yao et al., 2023). Since the reasoning process of o3 is not provided, we re-use o3 to generate concise intermediate reasoning content. These integrated trajectories with full reasoning process are used for SFT of our model.

**Tool Call Analysis.** We compare the distribution of tool calls per trajectory in our datasets against the datasets used in two representative works: ASearcher (Gao et al., 2025) and DeepDive (Lu et al., 2025). Since ASearcher only releases QA pairs, we generate the trajectories with o3. For a fair comparison, we randomly sample 800 examples from each dataset. As shown in Figure 3, three datasets have distinct patterns. ASearcher exhibits a highly skewed distribution, with a large proportion of examples requiring zero or very few tool calls (mean: 5.4, median: 1). DeepDive shows denser distribution of tool calls (mean: 9.5, median: 8), where the tool call number of most trajectories ranges from 6 to 20. In contrast, our dataset demonstrates substantially higher tool usage (mean: 20.3) with a broader distribution, featuring significant density in the 20–50 call range and non-negligible presence beyond 100 calls. This diverse and extended tool call distribution helps to enhance the robustness and performance of trained agents, particularly on benchmarks emphasizing sustained reasoning and tool calls.

## 3.2 CUSTOMIZED SEARCH TOOL DESIGN

The application of LLM Agents is not only dependent on the intelligence of the LLM, but also limited to the quality of the tool. For agentic RL training, efficiency and stability are significantly affected by the availability, throughput, and consistency of tools. In pursuit of throughput, many works (Jin et al., 2025; Chen et al., 2025b) deploy a naive search tool that retrieves documents from dumped wiki database based on vector search algorithms. In later experiments, we show that the quality of documents retrieved by this tool is limited and bounds the performance of the model. Other works (Li et al., 2025b; Lu et al., 2025) turn to paid commercial search services, which are confronted with problems such as expensive costs, privacy leakage. To solve these problems, we develop a dedicated in-house search and retrieval infrastructure, which provides scalable tool usage with high quality and throughput, enhancing transparency and facilitating further advancement.

Our search tool provides two functions, (1) Search: Given a query, the search function returns a batch of related URLs and snippets of the web pages. (2) Browse: Given a URL, the browse function returns a detailed document of the web page. Figure 4 illustrates the workflow of our tool.

**Search Function.** Given a query $q$, we first retrieve the search results $\{r_1, r_2, \ldots, r_n\}$ from a search engine, where $r_i = (s_i, t_i, url_i)$ consists of a snapshot $s_i$, title $t_i$ and the URL $url_i$ of a web page. The snapshot is a truncated sentence about the query within about 20 words, selected from the web page by the search engine. Though the snapshot is related to the query, it usually lacks enough information for the query. To improve the quality of retrieved documents and reduce redundant tool calls of the agent, we propose to manually generate a concise snippet with more compact information, without losing much efficiency. Specifically, we retrieve the full web page $p_i$ of $url_i$ via a web crawler, and then split $p_i$ into several short text chunks $\{c_1, c_2, \ldots, c_m\}$.

However, due to the web anti-crawler policy, the main texts of nearly 15% websites are unavailable to fetch. To solve this problem, we also add the snapshot into the chunks and obtain $C_i = \{c_1, c_2, \ldots, c_m, s_i\}$, so that $C_i$ contains at least one related chunk. Subsequently, we need to retrieve the most relevant chunks about the query. To improve the recall rate, it is common to generate multiple queries from different views in RAG systems (Li et al., 2024). This method is very effective but also time-consuming to generate multiple queries. Thus, we use the existing snapshot $s_i$ as another query, together with the original query $q$, to retrieve target chunks $C_i^t$:

$$C_i^t = \text{concat}(\text{BM25}_{k_q}(q, C_i), \text{BM25}_{k_s}(s_i, C_i)), \tag{5}$$

where $\text{BM25}_k(q, C)$ means retrieve the top-k relevant chunks about $q$ from corpus $C$ using the BM25 algorithm (Lù, 2024).

Then, we utilize an embedding model and a reranker model to further retrieve the most semantically relevant chunks $C_i^s$ from $C_i^t$. Finally, we prompt an LLM to generate a concise snippet $sn_i$ about $C_i^s$, focusing on $q_i$. In the end, the search function returns a list of search results $\{r_1^{sn}, r_2^{sn}, \ldots, r_n^{sn}\}$ to the agent, where $r_i^{sn} = (sn_i, t_i, url_i)$.

**Browse Function.** Given a $url$, the browse function returns the semantic document of the corresponding web page. This function endows the agent with the ability to seek more information deeply from interested documents, imitating human behaviors. We first retrieve the webpage $p$ from $url$, using our web crawler. Then $p$ is split into several long chunks ($> 2048$ tokens). To filter out noisy content, we select the webpage title as the query to retrieve the top-1 chunk $c$, using BM25, embedding model and reranker model as the retrieve method. Finally, the browse function returns $(url, c)$ to the agent.

## 4 EXPERIMENTS

### 4.1 EXPERIMENTAL SETUPS

**Benchmarks.** We evaluate InfoAgent on five public and challenging deep research benchmarks: BrowseComp (Wei et al., 2025), BrowseComp-ZH (Zhou et al., 2025b), Xbench-DeepSearch (Chen et al., 2025a)(Xbench-DS), SimpleQA (Wei et al., 2024), WebWalkerQA (Wu et al., 2025b).

**Baselines.** We compare InfoAgent against a diverse set of models, grouped into two main categories: (1) Proprietary Models: GPT-4o (OpenAI et al., 2024), o3 (OpenAI, 2025c), Claude-4-Sonnet (Antropic, 2025) and Kimi-Researcher (MoonshotAI, 2025). All models are equipped with search tools. (2) Open-Source Models: we compare against advanced models including GLM-4.5 (Zeng et al., 2025a), DeepSeek-V3.1 (DeepSeek-AI, 2025), WebDancer (Wu et al., 2025a), WebThinker (Li et al., 2025d), MiroThinker (MiroMindAI, 2025).

**Training Details.** For SFT, we fine-tune Qwen3-14B (Yang et al., 2025) on 14k synthesized trajectories using a batch size of 128 for 2 epochs. The learning rate is held at 2e-5 after 10 linear warmup steps. The context length is 32k tokens, and sequences exceeding this limit are truncated.

In the RL stage, we adapt GRPO (Shao et al., 2024) to further incentivize and enhance the capacity of the SFT model. The training dataset for RL is also synthesized by the method described in Section 3.1 but with more strict filtering strategies. Specifically, we calculate the $pass@4$ metric of the

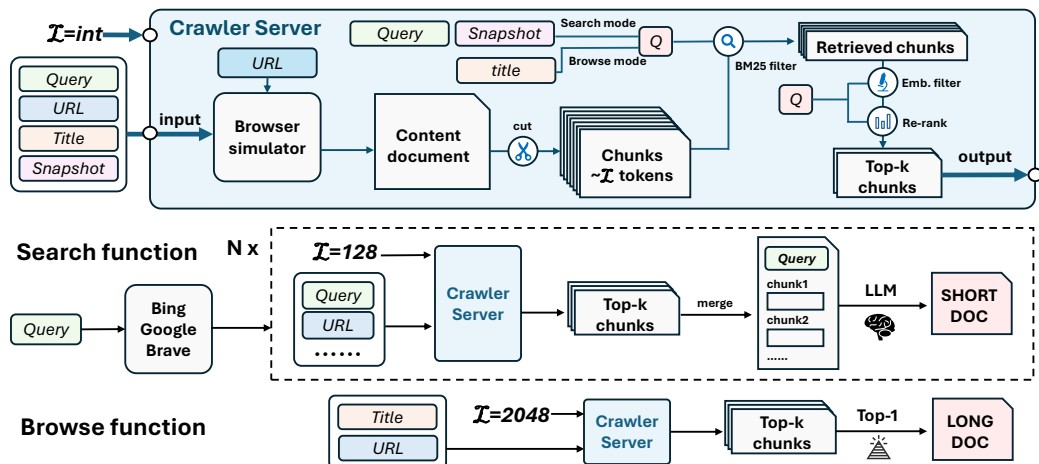

Figure 4: Workflow of our search tool. The search function receives a query and obtain initial results from the search engines. Then it crawls the corresponding web pages and extracts a concise document ($\leq$128 tokens) for each page, using BM25 filter, Embedding&Reranker Models, and LLM. The browse function receives a URL and extract a longer chunk ($\leq$2048 tokens) from the web page, using the same extraction pipeline with the search function, but without LLM for final generation.

SFT model on the dataset and then select the problems whose pass rate is between 0.25 and 0.75 under a 16k context length constraint, resulting in 5.7k samples. This strategy ensures the filtered dataset is of appropriate difficulty, which improves the efficiency and stability of RL training without the need for expansive dynamic oversampling during rollout (Yu et al., 2025). We use the AdamW optimizer with a constant learning rate of 1e-6 and linear warm-up over 60 steps. The batch size is set to 128 and the total epoch is set to 5. Since all the problems can be solved within 16k context length, we also truncate the response to 16k tokens during rollout, in order to speed up training and encourage the model to solve problems efficiently. We sample 8 trajectories in each group to compute the normalized advantage. The reward of each trajectory is set to 1 if the final answer is correct, otherwise the reward is 0.

## 4.2 MAIN RESULTS

**Dominance in the Small-size Open-Source Models.** The primary finding from our evaluation is that InfoAgent establishes a new state-of-the-art for open-source models in the sub-15B parameter class. As highlighted in the table, InfoAgent consistently outperforms all other models in its category across all benchmarks. Specifically, it achieves top scores of **15.3 on BrowseComp**, **29.2 on BrowseComp-ZH**, **40.4 on Xbench-DS**, and **52.7 on WebWalkerQA**. It is notable that our training data are all in English, the excellent performance of InfoAgent on Chinese benchmarks (BrowseComp-ZH, Xbench-DS) exhibits the cross-lingual generalization of our method. In addition, it is noteworthy that the final model after RL achieves substantial improvement over the SFT cold-start model, underscoring the effectiveness of our RL training pipeline.

**Competitive Performance Against Larger Models.** Notably, the performance of InfoAgent is not only dominant in its size class but is also highly competitive with, and in some cases surpasses, much larger open-source models (>15B). For example, on the BrowseComp benchmark, InfoAgent (15.3) outperforms larger models like WebSailor-72B (12.0) and MiroThinker-32B-DPO-v0.1 (13.0). Similarly, its score of **52.7 on WebWalkerQA** is higher than that of MiroThinker-32B (49.3) and WebThinker-32B (46.5).

**Comparison with Proprietary Models.** When benchmarked against leading proprietary models, InfoAgent demonstrates a compelling balance of performance and accessibility. While top-tier models like o3 and grok-4-fast maintain an edge on several benchmarks, InfoAgent achieves remarkable results. On the SimpleQA benchmark, InfoAgent (**90.4**) achieves performance that is nearly on par

Table 1: Evaluation on deep research benchmarks. Accuracy(%) is reported according to existing studies. **Bold** indicates the best performance among open-source models <15B, while underlined values represent the best performance among models <= 32B. The score of our models are computed as Avg@4. Our model achieves the best performance among open-source models < 15B, and even outperforms some larger models with 32B or 72B parameters.

| Model | BrowseComp | BrowseComp-ZH | Xbench-DS | SimpleQA | WebWalkerQA |
|---|---|---|---|---|---|
| *Proprietary Models* | | | | | |
| GPT-4o | 1.9 | 12.8 | 30.0 | 38.2 | 33.8 |
| o3 | 50.9 | 58.1 | 66.7 | - | 71.7 |
| OpenAI Deep Research | 51.5 | 42.9 | - | - | - |
| Gemini | - | - | 50+ | - | - |
| Perplexity | - | 22.6 | - | 93.9 | - |
| Grok-4-fast | 44.9 | 51.2 | 74.0 | 95.0 | - |
| Claude-4-Sonnet | 14.7 | 30.8 | 53.0 | - | 61.7 |
| Kimi-Researcher | - | - | 69.0 | 93.6 | - |
| *Open-Source Models* | | | | | |
| *>15B models* | | | | | |
| GLM-4.5 | 26.4 | 37.5 | 68 | - | 65.6 |
| DeepSeek-V3.1 | 30 | 49.2 | 71.2 | - | 61.2 |
| WebDancer-32B | 2.5 | 14.1 | 38.7 | - | - |
| WebThinker-32B | 2.8 | - | - | - | 46.5 |
| WebSailor-72B | 12.0 | 30.1 | 55.0 | 93.5 | - |
| WebSailor-32B | 10.5 | 25.5 | 53.3 | 92.8 | - |
| MiroThinker-32B-DPO-v0.1 | 13.0 | 17.0 | 41 | - | 49.3 |
| *<15B models* | | | | | |
| WebSailor-7B | 6.7 | 14.2 | 34.3 | - | - |
| DeepDive-9B | 6.3 | 15.1 | 38 | - | - |
| MiroThinker-14B-DPO-v0.1 | 9.0 | 11.1 | 30 | - | 47.2 |
| Qwen3-14B | 1.0 | 10.4 | 20.0 | 68.6 | 22.9 |
| InfoAgent-SFT (ours) | 4.7 | 17.0 | 28.0 | 80.3 | 41.5 |
| **InfoAgent (ours)** | **15.3** | **29.2** | **40.4** | **90.4** | **52.7** |

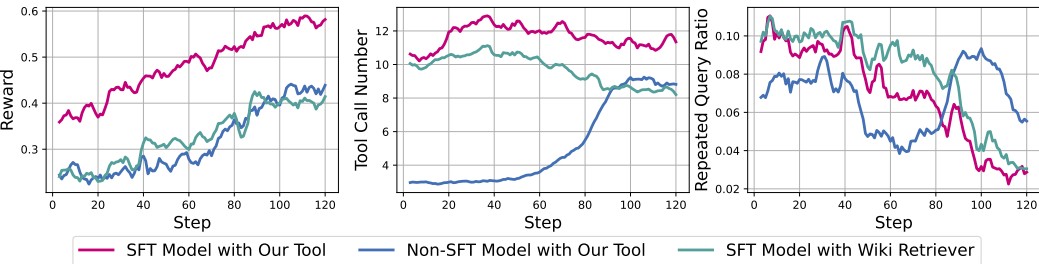

Figure 5: **Left:** Average reward (accuracy) per trajectory during the RL training process. **Middle:** Average tool calls per trajectory. Calls of search function and browse function are counted together. **Right:** Average ratio of repeated query per trajectory, which is the ratio of # unique queries and # all queries in the trajectory.

with large-scale models like WebSailor-32B (92.8) and the proprietary grok-4-fast (95.0), indicating a robust foundation in standard question-answering tasks.

## 4.3 ABLATIONS

**SFT Cold Start Before RL.** Many works (Guo et al., 2025; Zeng et al., 2025b) have discussed the impact of SFT cold starting before RL, especially in the domain of math reasoning. A popular opinion holds that if the model has already learned the necessary ability for a task from its pretrained data, then only SFT for formatting responses is needed. Unluckily, current base models commonly lack the ability for such new agent tasks as deep research. Figure 5 shows details of RL training with/without SFT cold starting. Compared with the SFT model, the initial instruct model hardly

Table 2: Comparison of accuracy and tool call number of models trained and evaluated with different settings across BrowseComp, BrowseComp-ZH and Xbench-DS. Models that skip SFT stage, or trained/evaluated with Wiki Retriever, performs worse.

| With SFT | Training Tool | Inference Tool | BrowseComp | | BrowseComp-ZH | | Xbench-DS | |
|---|---|---|---|---|---|---|---|---|
| | | | *Acc* | *Calls* | *Acc* | *Calls* | *Acc* | *Calls* |
| ✗ | Ours | Ours | 3.0 | 9.7 | 15.8 | 8.2 | 33.3 | 6.8 |
| ✓ | Ours | Ours | **10.0** | **33.5** | **18.7** | **20.4** | **35.3** | **16.5** |
| ✓ | Wiki Retriever | Wiki Retriever | 1.0 | 18.1 | 4.5 | 10.2 | 8.0 | 13.9 |
| ✓ | Ours | Wiki Retriever | 1.1 | 38.9 | 4.3 | 15.0 | 9.0 | 30.9 |
| ✓ | Wiki Retriever | Ours | 6.7 | 20.0 | 14.3 | 11.0 | 33.0 | 10.3 |
| ✓ | Ours | Ours | **10.0** | **33.5** | **18.7** | **20.4** | **35.3** | **16.5** |

Table 3: Impact of the number of tool calls in SFT trajectories on model performance. All models are evaluated with a 32k-token context limit and are forced to produce an answer. "OC" denotes the percentage of Out-of-Context cases.

| # Tool Calls | BrowseComp | | | BrowseComp-ZH | | | WebWalkerQA | | |
|---|---|---|---|---|---|---|---|---|---|
| | *Acc* | *Calls* | *OC* | *Acc* | *Calls* | *OC* | *Acc* | *Calls* | *OC* |
| $< 10$ | 2.8 | 8.8 | 5.0 | 5.9 | 4.0 | 3.0 | 36.2 | 5.6 | 2.0 |
| $\geq 10$ | 7.1 | 37.1 | 80.0 | 15.6 | 33.1 | 71.0 | 47.2 | 24.1 | 33.0 |

learns to gain more rewards and its accuracy has less improvement. Without SFT, the number of tool calls is also lower. Despite that the model learns to call tools more frequently, its ratio of repeated calls with the same query is also higher, indicating that the model does not know how to make full use of the tool. On the contrary, the SFT model learns to try different queries so that it can get more documents, and the repeat ratio clearly decays during the training progress. Since an agent for deep research needs a series of complex abilities such as planning, information retrieval, backtracking and so on, we argue that sufficient high-quality SFT is indispensable and directly impacts the effectiveness of further RL training.

**Quality of Search Tool.** We argue that the quality of the search tool is significant to model training and inference, especially for difficult and complex search tasks such as BrowseComp. For comparison with our well-designed tool, we deploy the wiki retriever used in Search-R1 (Jin et al., 2025), which is a simple tool retrieving documents from fixed wiki corpus, based on vector search. Since our dataset is also built from wiki, this retriever is able to provide necessary information for solving the problems. Figure 5 illustrates the training details using different tools. It is obvious that the accuracy is always higher when using our tool during RL process, proving that our tool can provide more useful information to help the agent tackle hard problems. Besides, the number of calls of wiki retriever is always lower than our tool, implying that the model cannot get higher accuracy if it calls more tools. Using wiki retriever, the model can only gain more reward from easy problems, and the accuracy finally converges to a low value. Benchmark results shown in Table 2 also demonstrate that using better tools in both training stage and inference stage can achieve best performance in downstream tasks, where the model can learn to call tools more frequently, resulting in higher accuracy. Note that, since the wiki retriever cannot access the web, it leads to consistently low performance on these open-web benchmarks, implying the limitations of traditional closed retrieval systems. In Section D.1, we further compare the effects of both tools on wiki-based benchmarks, which is a fair comparison that proves the advantage of our tool for model training. In summary, our well-designed search tool provides clean and concise documents which match the query perfectly. It is significantly beneficial to improve the upper bound of the model performance.

**Trajectory Length for SFT.** The number of tool calls within SFT trajectories has a substantial impact on model performance and behavior. To study this effect, we partition our synthesized trajectories into two subsets: those with fewer than 10 tool calls and those with at least 10 tool calls. We then train the base model on each subset and evaluate them on three benchmarks under a 32k-token context limit (see Table 3). Results show that models trained on longer trajectories ($\geq 10$ tool calls)

consistently achieve higher accuracy on all challenging benchmarks, but at the cost of excessive tool calling that often exhausts the available context. Conversely, models trained on shorter trajectories (<10 tool calls) fail to generalize well to complex tasks, as they tend to under-generate tool calls. Hence, in practice, we recommend incorporating a small proportion of short trajectories alongside longer ones to achieve strong performance while mitigating excessive tool calling.

## 5 CONCLUSION

In this paper, we present InfoAgent, an information-seeking deep research agent. To train this agentic model, we introduce a novel data synthesis pipeline based on sub-tree sampling with node obfuscation, which substantially increases query difficulty. On average, our synthesized queries require 20 tool calls for the OpenAI o3 model to answer correctly, significantly more challenging than datasets used in concurrent works such as Asearcher and DeepDive. In addition, we design a dedicated high-concurrency web search tool that enables effective and efficient model training. We show that InfoAgent achieves higher accuracy when equipped with this tool, compared with traditional tool. We evaluate the model across multiple benchmarks, including BrowseComp-en, BrowseComp-zh, WebWalkerQA, and Xbench-DS. Despite having only 14B parameters, InfoAgent matches or surpasses the performance of much larger open-source models, such as WebSailor-72B and DeepDive-32B.

A key limitation of our current work is that the context length is restricted to 16K tokens during reinforcement learning, which considerably constrains both the difficulty and the range of problems the model can address. This limitation can be mitigated by adopting base models with longer native context windows and leveraging more advanced reinforcement learning infrastructure. The data synthesis pipeline presented in this paper is evaluated on the Wikipedia corpus, which may constrain the scope of the synthesized problems. Expanding the pipeline to operate over the broader web holds promise for generating questions of even greater difficulty while simultaneously reducing the gap between training and real-world inference.

**Ethics Statement.** InfoAgent has the potential to improve research efficiency and knowledge discovery, but it also raises several ethical considerations. Our synthesized datasets are constructed from openly available Wikipedia content, which minimizes risks of privacy violations but still reflects biases and inaccuracies present in the source corpus. The model trained on these datasets may learn false facts and biases. From a societal perspective, autonomous research agents may also exacerbate misinformation risks if deployed without safeguards. We therefore encourage future work to incorporate human oversight, bias detection, and provenance tracking to ensure responsible and reliable use of such systems.

**Reproducibility Statement.** Given the synthesized dataset and our custom web search tool, InfoAgent can be reliably reproduced using standard supervised finetuning and GRPO reinforcement learning. The data synthesis pipeline can be faithfully reconstructed by following the procedures detailed in this paper, though the resulting data may not be identical due to inherent randomness. Reproducing the custom search tool presents greater challenges, as it demands substantial iterative engineering to reduce latency and optimize distributed parallelism.

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

# A DATA SYNTHESIS DETAILS

## A.1 SUBTREE EXTRACTION

---

**Algorithm 1** Subtree Extraction with same root

---

**Require:** Tree $T = (V, E)$ with parent pointers $\text{PARENT}(\cdot)$, budget $k$
**Ensure:** Subtree $T'$ containing all nodes
1: **function** EXTRACTER$(T, k)$
2:     $k \leftarrow \min(k, |V|)$                                                       $\triangleright$ cap budget by number of nodes
3:     $S \leftarrow \emptyset$                                                                $\triangleright$ selected node set
4:     **while** $k > 0$ **do**
5:         $n \leftarrow \text{RANDOM}(V)$
6:         $P \leftarrow \emptyset$                                         $\triangleright$ nodes on the path $n \to$ root via PARENT
7:         $u \leftarrow n$
8:         **while** $u \notin P$ **do**
9:             $P \leftarrow P \cup \{u\}$
10:            **if** $\text{PARENT}(u)$ is NONE **then**                 $\triangleright$ reached a root
11:                **break**
12:            **else**
13:                $u \leftarrow \text{PARENT}(u)$
14:         $\text{cost} \leftarrow |\{x \in P \mid x \notin S\}|$
15:         **if** $\text{cost} > 0 \wedge \text{cost} \le k$ **then**
16:            $S \leftarrow S \cup P$                                   $\triangleright$ take entire path
17:            $k \leftarrow k - \text{cost}$
18:     $T' \leftarrow \text{BUILDSUBTREE}(T, S)$
19:     **return** $T'$
20: **function** BUILDER$(T, S)$
21:     $V' \leftarrow S$
22:     $E' \leftarrow \{(u, v) \in E \mid u \in S \wedge v \in S\}$
23:     **return** $(V', E')$

---

## A.2 SFT REASONING CONTENT GENERATION

We prompt OpenAI o3 OpenAI (2025c) to generate reasoning for next tool call given previous reasoning, tool calls and tool responses. To encourage flexibility, we use a lightweight prompt and only restrict the length of the generated content to ensure concise reasoning.

> **Prompt for Reasoning Generation**
>
> Imagine you are a smart web agent that can use web search and web browse to get information online and answer user's question. You always think with reasoning, and then decide which tool to use and how to use it, turn by turn, until you reach a final answer. Now I have a conversation history of you and the user: conversation Notice that before your last message that contains tool calls, there's a message from you containing empty reasoning content, please fill it in. Please directly produce the content without any tags. Do not produce more than 5 sentences.

## B  DETAILS OF SEARCH TOOL

In this section, we provide more details about the implementation of our search tool, including hyper-parameter settings, real-time performance, and optimization mechanisms.

### B.1  SEARCH ENGINES

We primarily use the Google Search Web API to retrieve URLs for most requests, accounting for its high availability and quality. For the case where Google returns empty results, we fall back to Brave and Bing. The default region and language of the search engine is `us-en`. When evaluating the model on Chinese benchmarks, this is set to `cn-zh`.

### B.2  HYPER-PARAMETERS

For the search function, the maximum number of search results for a query returned to the model is set to 5. More results do not further improve the final performance but waste tokens. The web page is split into chunks with the size of 128 tokens. Then we use the BM25 to retrieve the top-40 chunks, and use the snapshot as the query to retrieve another top-3 chunks. The snapshot itself, as the seed chunk, is added to the final chunks. Later, we use Qwen-3-Embedding-0.6B to retrieve top-8 chunks from these 44 chunks, and then use Qwen-3-Reranker-0.6B to retrieve top-3 chunks from these 8 chunks. Finally, we combine these chunks with the original query and send to GPT-4o-mini for snippet generation.

For the browse function, we split the web page into chunks with the size of 2048 tokens, and retrieve the top-40 chunks using BM25, then top-8 chunks using the embedding model, and finally we get the top-1 chunk using the re-rank model.

---

**Prompt for Snippet Generation**

I will give you a part of content in a html file, and a query. Complete the following task: <task description>According to the given content, generate a concise snippet. The snippet should be within 60 words. If the content has information about the query, the snippet should focus on the information about the query and try to answer the query. Else if the content has no information about the query, IGNORE the query and the snippet should just be a concise summary of the given content without mentioning the query. IGNORE irrelevant information commonly found in websites such as advertisement, navigation bar, cookies, login notice, etc. Also give the relevance of the snippet to the query, range from 0 to 1. If there is no related information about the query in the content, set the relevance to 0 and set the snippet to a concise summary of the given content. REMEMBER: Do not include any thinking or explanation in the snippet, the snippet must be the final summary of the given content, and must be within 60 words. Do not include your own knowledge, all the information in the snippet must be from the given content. You MUST NOT say the content does not mention the query, I don't need you to notice me, just give a concise summary of the content. Again, if the content has no information about the query, DO NOT mention that in the snippet. Instead, provide a concise summary of the given content as the snippet, just like the query is never given. Ignore contents unrelevant to the main text, such as cookies, login information, privacy statements of the web page. Return the result in the following JSON format: {'snippet': '...', 'relevance': ...}.</task description>\n Here is the content and query: \n<content>\n{text}\n</content>\n<query>\n{query}\n</query>

---

### B.3  PERFORMANCE OPTIMIZATION

Throughput and latency of the environment feedback is one of the main bottlenecks of agentic RL training. The cost of pursuing high quality of retrieved documents is high latency. Though the QPS of tool calling can be improved by scaling up the CPU clusters, lowering the latency of each request remains a challenge.

To alleviate this problem, we deploy a Redis server to save all intermediate results of each request. When processing a new request, we first try to read the results from caches. This cache mechanism

is very effective in closed scenarios such as RL training via GRPO, where a group of trajectories solving the same problem often share the majority of queries and the hit rate of caches is commonly high. In such cases, average latency can be dramatically decreased by the cache mechanism.

Under our current computational resources and engineering optimizations, the tool achieves a throughput of approximately 50 queries per second (QPS) with an average latency of 6–7 seconds. In our RL rollout setting, the average tool call rate is around 30 QPS, which the system comfortably supports. Because model generation and tool calls are fully asynchronous, the total time for a rollout is roughly determined by the model generation time of the longest trajectory, plus the number of tool calls multiplied by the latency. As long as latency is sufficiently low to maintain full GPU utilization, the practical bottleneck of our system lies in achievable QPS rather than individual query latency.

### B.4 ERROR-HANDLING MECHANISM DURING TRAINING

Our training framework (verl (Sheng et al., 2024)) returns the error message to the model whenever a tool call fails. Failures may arise from two sources:

- Tool-side errors (e.g., code issues, network timeouts). We have devoted substantial engineering effort to ensuring the high availability of the self-hosted tool. In practice, it provides a stable service during training. A small proportion of failed calls is acceptable in reinforcement learning training and has minimal impact on stability, as these failures effectively serve as part of the environment's feedback rather than numerical noise.

- Model-side errors (e.g., incorrect call formatting). In such cases, the framework passes the error message back to the model, allowing it to continue generation and learn from its mistakes. Since repeated failed calls reduce the final reward, the model is incentivized to adjust and produce correct tool calls over time. As an alternative, the framework can also assign a zero reward to the failed turn and terminate generation, achieving a similar training effect.

### B.5 ABLATION OF SNIPPET GENERATOR

In our implementation, we use GPT-4o-mini to generate the final snippet, which is essentially a simple summarization task. Table 4 illustrates that using models with different sizes for snippet generation makes little difference to the final performance. Directly using the top-1 chunk returned by the reranker can lead to information loss and worse performance. While using the top-3 chunks consumes tokens fastly, leading the model achieving maximum context limit (64k) without sufficient search.

Table 4: Performance of InfoAgent on BrowseComp using different snippet generator. Rerank top-$k$ means using the top-$k$ chunks returned by the reranker as the snippet.

| Snippet Generator | Acc | Avg. Token |
|---|---|---|
| GPT-4o-mini | 15.3 | 53k |
| Qwen3-4B | 15.2 | 52k |
| Qwen3-0.6B | 14.7 | 53k |
| Rerank top-1 | 13.4 | 57k |
| Rerank top-3 | 9.7 | 60k |

## C MORE DETAILS ABOUT TRAINING AND EVALUATION

### C.1 REINFORCEMENT LEARNING ALGORITHM

In the RL stage, we adapt GRPO Shao et al. (2024) to further incentivize and enhance the search ability that the model has learned during SFT stage. Specifically, for each question $q$ and its ground-truth answer $a$ from a dataset $D$, the old policy $\pi_{\theta_{old}}$ generates a group of trajectories $\{o_1, o2, \ldots, o_G\}$,

and then we optimize the policy $\pi_\theta$ by maximizing the following objective:

$$J_{\text{GRPO}}(\theta) = \mathbb{E}_{(q,a)\sim D,\ \{o_i\}_{i=1}^G \sim \pi_{\theta_{\text{old}}}(\cdot|q)}$$

$$\left[ \frac{1}{G} \sum_{i=1}^G \frac{1}{|o_i|} \sum_t^{|o_i|} \left( \min\left[ R_{i,t}(q)A_{i,t}, \text{clip}(R_{i,t}(q), 1-\epsilon, 1+\epsilon)A_{i,t} \right] - \beta D_{KL}(\pi_\theta||\pi_{\text{ref}}) \right) \right], \quad (6)$$

where $R_{i,t}(q) = \frac{\pi_\theta(o_{i,t}|q,o_{i,<t})}{\pi_{\theta_{\text{old}}}(o_{i,t}|q,o_{i,<t})}$ is the importance sampling ratio, $\epsilon$ and $\beta$ are hyper-parameters that control the clipping threshold of $R_{i,t}(q)$ and the weight of KL penalty term. In our experiments, $\beta$ is set to 0, in order to encourage the model to explore new strategies. The advantage $A_{i,t}$ is estimated by the normalized reward of the trajectory within the group:

$$A_{i,t} = \frac{r_i - \text{mean}(\{r_1, r_2, \ldots, r_G\})}{\text{std}(\{r_1, r_2, \ldots, r_G\})}. \quad (7)$$

Here, $r_i$ is the reward assigned to the trajectory $o_i$. Following previous work Wei et al. (2025), we utilize LLM-as-judge to compute the correctness score of the trajectory as the final reward:

$$r_{i,\text{correctness}} = \begin{cases} 1, & \text{if response equals to ground truth answer,} \\ 0, & \text{else.} \end{cases}$$

## C.2 Evaluation Details

We use the same method described in BrowseComp Wei et al. (2025) to verify the correctness of an answer from the model. Specifically, given a question, the ground truth and the response of the model, we use o4-mini OpenAI (2025c) to extract the final answer to the question from the response of the last turn and verify its correctness according to the ground truth.

The maximum context length is set to 64k by default. If the model continues calling tools when reaching the maximum context length, we force the model to output the final answer.

---

**Prompt for answer verification**

You are an evaluator. Based ONLY on the `[correct_answer]`, judge whether the `[response]` to the `[question]` is correct.
=== INPUTS ===
`[question]`: {question}
`[response]`: {answer}
`[correct_answer]`: {ground_truth}
=== TASK ===
1. Extract the single final answer from the `[response]`. If no clear final answer exists, write "None".
2. Give a concise explanation (reasoning) that ONLY compares the extracted answer with the `[correct_answer]`. Do not solve the problem again or add extra background.
3. Decide correctness: set correctness = correct if they are equivalent / within a tiny numeric tolerance and acceptable difference of expression style; otherwise incorrect. `[correct_answer]` may contain multiple answers separated by "OR", the response is correct if it matches any of the answers.
4. Extract a confidence score (0–100). If the `[response]` provides none, use 100.
=== OUTPUT FORMAT (STRICT) ===
Return a valid JSON object with exactly these keys:
{
"extracted_final_answer": <string>,
"reasoning": <string>,
"correctness": <string "correct" or "incorrect">,
"confidence": <integer 0-100>
}
Do NOT output anything else—no comments, no code fences.

---

## C.3 Training Details of Ablation Experiments

In Sec.4.3, we train 3 models with different settings. They share most training settings with our main experiments as described in Sec.4.1, except that we chose their checkpoints at the 120th step in RL stage for comparison and evaluation, considering the time limit. The maximum context length is also set to 32k to speed up evaluation. The Wiki Retriever is the same as the one used in Search-R1 Jin et al. (2025), which uses e5-base-v2 Wang et al. (2022) to encode texts and retrieve documents from the dumped database of Wiki-18 via vector search. The maximum number of retrieved documents of our tool and Wiki Retriever is set to 5. For fairness, we only alter the search function to the Wiki Retriever, and the browse function is kept the same.

# D More Ablation Experiments

## D.1 Effects of Different Tools on Wiki-based Benchmarks

Since the wiki retriever accesses only a fixed offline Wikipedia dump, it leads to extremely low inference performance on our main open-web benchmarks, as discussed in Section 4.3. For fair comparison, we extend this ablation on several widely used wiki-based benchmarks, where the wiki retriever can be properly applied, including NQ (Kwiatkowski et al., 2019), TriviaQA (Joshi et al., 2017), PopQA (Mallen et al., 2023), HotpotQA (Yang et al., 2018), Musique (Trivedi et al., 2022) and Bamboogle (Press et al., 2023).

Table 5: Comparison of accuracy of models trained and evaluated with different tools on wiki-based benchmarks. Models trained with our tools performs consistently better than models trained with the wiki retriever.

| Training Tool | Inference Tool | Bamboogle | PopQA | NQ | TriviaQA | HotpotQA | Musique |
|---|---|---|---|---|---|---|---|
| Wiki Retriever | Wiki Retriever | 66.4 | 55.9 | 64.8 | 82.8 | 67.9 | 31.3 |
| Ours | Wiki Retriever | **70.4** | **56.6** | **66.1** | **83.6** | **68.9** | **33.1** |
| Wiki Retriever | Ours | 79.2 | 58.0 | 65.3 | 89.0 | 70.1 | 31.7 |
| Ours | Ours | **86.4** | **59.7** | **67.2** | **90.3** | **72.7** | **34.9** |

As shown in Table 5, when inference is performed with the same tool, the model trained using our tool consistently outperforms the model trained directly with the wiki retriever across all these datasets. This suggests that our tool provides richer training dynamics, leading to more effective reasoning strategies.

## D.2 Effects of SFT Cold Start on the $pass@k$ Metric of the Model

Table 1 illustrates that RL brings a dramatic improvement to the SFT model, which raises its accuracy on BrowseComp from 4.7% to 15.3%. While Table 2 shows that RL brings limited improvement without SFT cold start. To elaborate on this phenomenon, we measure the $pass@k$ accuracy of the SFT model, based on a recent theory that RL improves sampling efficiency—effectively shifting its $pass@1$ performance closer to its $pass@k$ performance (Yue et al., 2025). Given a problem, we sample $k$ outputs from the model. The $pass@k$ value for this problem is 1 if at least one of the $k$ samples passes verification; otherwise, it is 0. The average $pass@k$ value over the dataset reflects the proportion of problems in the dataset that the model can solve within $k$ trials, providing a rigorous evaluation of the reasoning capacity coverage of LLMs.

Table 6: Comparison of $pass@k$ accuracy on BrowseComp between the initial model (Qwen3-14B) and the SFT model (InfoAgent-SFT). The SFT model shows dramatic improvement, implying its boosted potential.

| Model | $pass@1$ | $pass@4$ | $pass@8$ | $pass@16$ |
|---|---|---|---|---|
| Qwen3-14B | 1.0 | 4.3 | 6.0 | 8.0 |
| InfoAgent-SFT | 4.7 | 19.4 | 28.7 | 37.3 |

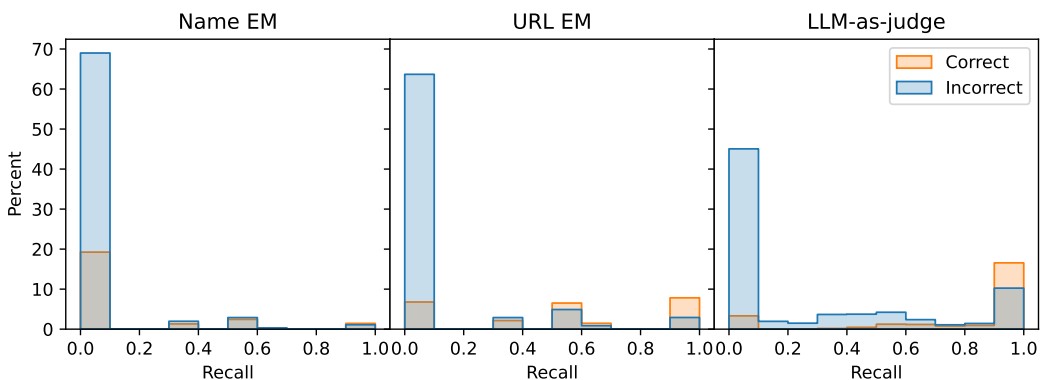

Figure 6: Distribution of recall rates calculated by different methods

As shown in Table 6, the SFT model (InfoAgent-SFT) achieves a $pass@4$ accuracy of 19.3%, compared to 4.3% for the base model (Qwen3-14B-Instruct). This suggests that the SFT cold start enables the model to encode substantive strategic knowledge, though its $pass@1$ performance (4.7%) under-utilizes that capacity. The result of the RL-trained InfoAgent (15.3%) remains below this theoretically possible performance, indicating that RL mainly helps the model better express capabilities already learned during SFT.

## E  REINFORCEMENT LEARNING WITH PROCESS REWARD

Since reward is the only external training signal in reinforcement learning, it directly impacts the final performance of the model. Following previous work Wei et al. (2025), we utilize the evaluation method described in Sec.C.1 to compute the score of the model response as the final reward.

This outcome-based binary reward has been proved effective by many researches Guo et al. (2025); Yu et al. (2025). Some works Luo et al. (2024); Lightman et al. (2023) also argue that intermediate process reward is beneficial, while it is commonly hard to obtain process rewards for many tasks.

Thanks to our structured construction process of search problems, intermediate artifacts produced during data construction can provide additional process rewards for the trajectory. This enables us to investigate whether process rewards help to improve training efficiency and model performance.

For each question in the dataset, we store the involved entities and the corresponding source webpage URL. Since a question can only be solved after all the entities are found, we compute the recall rate of target entities in the trajectory as the additional bonus added to the final reward. There are three methods to determine whether an entity is recalled in a trajectory: (1) Name EM: search the entity name via Exact Match, (2) URL EM: search the source URL of the entity via Exact Match, (3) LLM-as-judge: determine the existence of the entity via LLM. Here we use Qwen3-Reranker-0.6B as the judge, which is good at determining whether a document is related to a given query. Specifically, we use the retrieved snippet and web page in the trajectory as the corpus, and use the entity name as the query. The re-rank score between the query and the top-1 document will be close to 1 if the entity exists, else 0. Middle values seldom occur. Thus we use this score to calculate the recall rate.

To estimate how much bonus the model can get during training, we sample its trajectories on the training dataset and calculate the recall rates via the three methods. Fig.6 shows the distributions of the recall rates. Obviously, using LLM-as-judge can get more non-zero recall bonus. Thus, we apply this method to calculate additional process reward $r_{i,\text{recall}}$ for trajectories that failed to find the correct final answer. Specifically, the new reward function is defined as

$$r_i = \max(r_{i,\text{correctness}} + \lambda r_{i,\text{recall}}, 1) \tag{8}$$

where $\lambda$ controls the weight of the recall bonus. The max function ensures that a trajectory can get the highest score if and only if its final answer is correct, preventing potential reward hacking.

Table 7: Benchmark Performance with different $\lambda$ value

| $\lambda$ | BrowseComp | BrowseComp-ZH | Xbench-DS |
|-----------|------------|---------------|-----------|
| 0 | 15.3 | 29.2 | 40.9 |
| 0.5 | 15.1 | 26.8 | 39.6 |
| 0.9 | 14.5 | 28.0 | 37.5 |

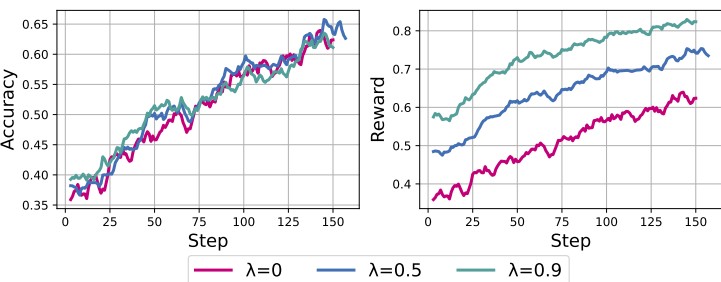

Figure 7: Training Curve with different weight of bonus

Fig.7 illustrates the training dynamics with different $\lambda$. Obviously, the reward curve is lifted as we increase $\lambda$. When $\lambda = 0.9$, the reward curve keeps about 0.2 higher than when $\lambda = 0$, which means there are a number of failed trajectories that gain the recall bonus. However, there is no significant difference in the accuracy curve. Additionally, Tab.7 also demonstrates that increasing $\lambda$ does not help improve the final performance of the model.

We propose two assumptions to explain this result. (1) The recall bonus is added to the final reward, and the exact step that finds the target entity does not obtain clear positive feedback. This makes it difficult for the model to learn from these trajectories. (2) The training dataset is relatively simple, and the correctness reward is enough to teach the model most of the necessary skills. We leave the specific mechanism of process reward for agentic RL training to future work.

## F  CASE STUDY

To illustrate how our agent framework operates in practice, we provide a detailed trajectory example generated during RL rollout. This example demonstrates the agent's multi-step reasoning process, thought, tool call, and tool response when handling complex queries.

---

**Question:** With which Major League Baseball team did the pitcher—born in the 1930s in the Virginia county that Congress returned to its state in 1846 and renamed in 1920, and who ranked third in the league after recording nearly twenty saves in a mid-century season—make his major-league debut?

**Ground Truth:** the Cleveland Indians

---

**STEP 1**

**Thought**

```
<think>
```
The user is asking for the MLB team with which a specific pitcher made his debut.
Key clues to identify the pitcher:
1. Born in the 1930s.
2. Birthplace: a Virginia county that Congress returned to the state in 1846 and renamed in 1920.
3. In a mid-century season he recorded "nearly twenty" saves and finished third in the league in saves.
First I need to determine which Virginia county matches the 1846 return and 1920 renaming. Once I have the county, I can look for pitchers born there in the 1930s who had a notable save season. Then I can find the team with which he debuted.
I'll start by searching for Virginia counties that were returned to Virginia in 1846 and renamed in 1920.
```
</think>
```

**Action**

```
<tool_call>
{"name": "search", "arguments": {"query": "Virginia county returned to Virginia in 1846 re-
named in 1920"}}
</tool_call>
```

**Observation**

```
<tool_response>
```
{"url": "https://en.wikipedia.org/wiki/Arlington_County,_Virginia", "title": "Arlington County, Virginia", "snippets": "Alexandria County was retroceded back to Virginia in 1847 after a referendum showed majority support.  The area, originally ceded to form the District of Columbia in 1801, faced economic challenges that led to this decision.  The county was renamed in 1920, reflecting its historical significance and connection to Virginia."}, {"url": "https://en.wikipedia.org/wiki/Alexandria,_Virginia", "title": "Alexandria, Virginia", "snippets": "On July 9, 1846, Congress retroceded Alexandria County to Virginia.  The remaining portion of Alexandria County was renamed Arlington County in 1920.  The City of Alexandria became independent of Alexandria County in 1870 after being re-chartered in 1852."}, {"url": "https://arlhist.org/1920s-arlington-in-a-dozen-objects/", "title": "1920s Arlington in a Dozen Objects", "snippets": "In March 1920, Alexandria County was officially renamed Arlington County. It had been called Alexandria County since 1846 when it was retroceded back to Virginia, leading to confusion with Alexandria City.  The renaming aimed to clarify address identification in the area."}, ...
```
</tool_response>
```

---

## STEP 2

**Thought**

```
<think>
```
The search results confirm that the county in question is Arlington County, Virginia (formerly Alexandria County, retroceded in 1846 and renamed in 1920). Therefore, I need to find a pitcher born in Arlington County in the 1930s who, in a mid-century season, . . .
```
</think>
```

**Action**

```
<tool_call>
{"name": "search", "arguments": {"query": "born in Arlington County Virginia pitcher 1930s"}}
</tool_call>
```

**Observation**

```
<tool_response>
{"url": "https://en.wikipedia.org/wiki/List_of_people_from_Virginia", "title": "List of people from Virginia", "snippets": "Billy Wagner (born 1971) is a retired MLB pitcher known for his time with the Houston Astros and New York Mets, as well as pitching for Ferrum College. He is recognized for his contributions to baseball, but there is no mention of a pitcher from the 1930s in the provided content."}, . . .
. . .
</tool_response>
```

## STEP 3

**Thought**

```
<think>
```
The initial search for "born in Arlington County Virginia pitcher 1930s" returned only broad lists of Virginia-born players and no obvious matches. To zero in on the specific individual, I should incorporate the distinctive clue about his save statistics: he finished third in the league in saves in a mid-century season after recording "nearly twenty" saves. . . .
```
</think>
```

**Action**

```
<tool_call>
{"name": "search", "arguments": {"query": "third in saves 1950s nearly twenty saves"}}
</tool_call>
```

**Observation**

```
<tool_response>
{"url": "https://en.wikipedia.org/wiki/Roy_Face", "title": "Roy Face", "snippets": "Face was the first major leaguer to save 20 games more than once, leading the league three times. In 1959, he set a major league record for winning percentage and single-season wins in relief. He also held the NL record for career saves with 193 until 1982, showcasing his dominance in the 1950s."}, . . .
. . .
</tool_response>
```

## STEP 4

**Thought**

```
<think>
```
The previous search results were dominated by references to Roy Face and Hoyt Wilhelm, who were born in Pennsylvania and New Jersey, respectively, so they clearly do not match the Arlington County birthplace clue. I need to narrow the query to pitchers who were actually born in Arlington County, . . .
```
</think>
```

**Action**

```
<tool_call>
{"name": "search", "arguments": {"query": "Arlington County Virginia born baseball pitcher"}}
</tool_call>
```

**Observation**

```
<tool_response>
{"url": "https://en.wikipedia.org/wiki/Sean_Doolittle", "title": "Sean Doolittle", "snippets":
"Sean Robert Doolittle, born September 26, 1986, is a former professional baseball pitcher from
Virginia. He played for the University of Virginia, where he set a record for career wins as a
pitcher with 22. Doolittle was also recognized as the Atlantic Coast Conference Player of the
Year in 2006."}, . . .
</tool_response>
```

## STEP 5

**Thought**

```
<think>
```
The previous search for "Arlington County Virginia born baseball pitcher" mostly surfaced modern players such as Sean Doolittle, who clearly do not fit the 1930s birth-year clue. To zero in on the correct era, I should explicitly include a 1930s birth year in the query. . . .
```
</think>
```

**Action**

```
<tool_call>
{"name": "search", "arguments": {"query": "born 1930 Arlington Virginia baseball"}}
</tool_call>
```

**Observation**

```
<tool_response>
{"url": "https://en.wikipedia.org/wiki/Bill_Dailey", "title": "Bill Dailey", "snippets": "William
Garland Dailey (May 13, 1935 – January 11, 2025) was an American Major League Baseball
relief pitcher who played in all or part of four seasons. He was born in Arlington County, Virginia,
and played for teams including the Cleveland Indians and Minnesota Twins. Dailey passed away
in Bedford, Virginia, at the age of 89."}, . . .
. . .
</tool_response>
```



**STEP 6**



**Thought**

```
<think>
```
The search results list William Garland "Bill" Dailey, a relief pitcher born in Arlington County, Virginia, on May 13, 1935. That matches the birthplace and 1930s birth-year clues. I now need to confirm that he had a mid-century season in which he recorded "nearly twenty" saves and finished third in the league, and then determine the team with which he made his major-league debut. I'll open his Wikipedia page to verify these details.
```
</think>
```

**Action**

```
<tool_call>
{"name": "browse", "arguments": {"url": "https://en.wikipedia.org/wiki/Bill_Dailey"}}
</tool_call>
```

**Observation**

```
<tool_response>
```
{"url": "https://en.wikipedia.org/wiki/Bill_Dailey", "semanticDocument": "Bill Dailey \n| Bill Dailey | |\n|—|—|\n| Pitcher | |\n| Born: (1935-05-13)May 13, 1935 Arlington, Virginia, U.S. | |\n| Died: January 11, 2025(2025-01-11) (aged 89) Bedford, Virginia, U.S. | |\n Batted: Right Threw: Right | |\n| MLB debut | |\n| August 17, 1961, for the Cleveland Indians | |\n| Last MLB appearance | |\n| June 9, 1964, for the Minnesota Twins | |\n| MLB statistics | |\n| Win–loss record | 10–7 |\n| Earned run average | 2.76 |\n| Strikeouts | 109 |\n| Saves | 22 |\n| Stats at Baseball Reference | |\n| Teams | |\n William Garland Dailey (May 13, 1935 – January 11, 2025) was an American Major League Baseball relief pitcher who played in all or part of four seasons for the Minnesota Twins from 1963 to 1964. The right-hander stood 6 feet 3 inches (1.91 m) tall and weighed 185 pounds (84 kg). He was born in\n Dailey spent almost nine full years in minor league baseball before his recall by the Indians in August 1961. In Dailey's one full Major League season—1963 with the Twins—he appeared in 66 games (second among American League pitchers), compiling a 6–3 record and a 1.99 earned run average. His 21 saves ranked third in the league that season. But he sustained a rotator cuff injury in 1964, and he retired from baseball after the season.[1]\n In 119 career MLB games pitched, all in relief, Dailey posted a 10–7 won–lost record, 22 saves, and a 2.76 earned run average. In 185⅔ innings pitched, he struck out 109, allowing 162 hits and 59 bases on balls.\n Dailey died in Bedford, Virginia on January 11, 2025, at the age of 89. . . .
```
</tool_response>
```



**STEP 7**



**Thought**

```
<think>
```
The Wikipedia article confirms that the pitcher is William Garland "Bill" Dailey: born 13 May 1935 in Arlington, Virginia (the county retroceded in 1846 and renamed in 1920). In 1963 he recorded 21 saves—"nearly twenty"—and finished third in the American League in saves. The article also states that his major-league debut came on 17 August 1961 with the Cleveland Indians. Therefore, the team with which he made his debut was the Cleveland Indians.
```
</think>
```
He made his major-league debut with the Cleveland Indians.

## G USE OF LLM

In the preparation of this manuscript, Large Language Models (LLMs) were utilized solely for the purpose of refining the writing. Specifically, GPT-4o was employed as an editorial assistant to improve grammar, enhance sentence structure, and increase the overall clarity and readability of the text.

All scientific content, original ideas, arguments, data analysis, and conclusions presented in this paper were conceived, developed, and written entirely by the human authors. The LLM's role was strictly limited to stylistic improvements and linguistic polishing, operating under the direct supervision and final approval of the authors.

