# OpenReview forum: "InfoAgent: Advancing Autonomous Information‑Seeking Agents"
_ICLR.cc/2026/Conference — Submitted to ICLR 2026_

### Official Review · Reviewer_7jmx · 2025-10-28

**Soundness:** 2
**Presentation:** 3
**Contribution:** 2
**Rating:** 4
**Confidence:** 4

**Summary:**

This paper introduces InfoAgent, a 14B-parameter deep research agent trained via supervised fine-tuning (SFT) and reinforcement learning (RL) to perform multi-hop web search and information retrieval. The authors make three main contributions: (1) a data synthesis pipeline that constructs challenging multi-entity questions through entity tree construction, fuzzification, and sub-tree sampling; (2) a custom self-hosted search infrastructure with enhanced snippet generation to replace commercial APIs; and (3) a two-stage training recipe (SFT + GRPO) applied to Qwen3-14B. InfoAgent achieves 15.3% accuracy on BrowseComp, 29.2% on BrowseComp-ZH, and 40.4% on Xbench-DS, establishing state-of-the-art performance among open-source models under 15B parameters.

**Strengths:**

1. **Originality in data synthesis**: The entity tree construction with three-stage fuzzification (entity names, dates/numbers, semantic rephrasing) is a creative approach to systematically increase question difficulty.
2. **Quality of engineering**: The custom search infrastructure with multi-stage retrieval (BM25 -> embedding -> reranker -> LLM snippet generation) demonstrates good systems engineering. The Redis caching and performance optimizations show attention to practical RL training requirements.
3. **Thorough empirical analysis**: The ablation studies provide valuable insights:
   - SFT cold-start is essential (Figure 5 shows non-SFT model fails to learn)
   - Tool quality significantly impacts performance (Table 2)
   - Trajectory length matters (Table 3)
   - Process rewards don't help (Appendix D)

**Weaknesses:**

1. **Missing critical baselines**: The paper cites ChatGPT Deep Research (OpenAI, 2025a), Gemini Deep Research (Google, 2025), Perplexity Deep Research (Perplexity, 2025), and Grok Deep Search (xAI, 2025a) as motivating examples but does not evaluate against any of them. This is a critical omission that makes it impossible to assess the true contribution relative to deployed deep research systems.
2. **Limited performance and overstated claims**: The title claims to "advance" autonomous information-seeking agents, but InfoAgent achieves only 30% of o3's performance and lags significantly behind DeepSeek-V3.1 (71.2% vs 40.4% on Xbench-DS) and GLM-4.5 (68% vs 40.4%). The contribution is better characterized as advancing *small* open-source agents, not the field overall.
3. **Limited novelty**: The paper primarily combines existing techniques (ReAct framework, GRPO, entity-based QA generation, multi-stage retrieval). While the integration is competent, there is insufficient algorithmic innovation.

**Questions:**

1. **GLM-4.5 and DeepSeek-V3.1 setup**: Were these large open-source models evaluated with search capabilities? If so, what search infrastructure did they use? This affects interpretation of whether InfoAgent's contribution is the model training or the search tool.

---

> ### Author Response · Authors · 2025-11-20
>
> We thank the reviewer for your thorough reviews and insightful comments on our paper. We address your concerns as follows:
>
> > ### *W1: Missing critical baselines: The paper cites ChatGPT Deep Research, Gemini Deep Research, Perplexity Deep Research, and Grok Deep Search as motivating examples but does not evaluate against any of them.*
>
> The industrial frontier models are a lot more competent than our agent, due to scaling and other technical innovations. We treat them as motivating examples to investigate the undisclosed training methodologies. We fully respect these frontier models for leading this important research direction, but our current model with 14B size and 16K context is not directly comparable with these models. In the paper, we have already listed the performance of o3, Claude-4-Sonnet and Kimi-Researcher. And because they are just for reference instead of baselines, we do not list all commercial deep research agents.
>
> > ### *W2: The contribution is better characterized as advancing small open-source agents, not the field overall.*
>
> Thank you for this valuable feedback. We fully acknowledge that our current results remain below that of larger models such as DeepSeek-V3.1 and GLM-4.5. Our focus is not on outperforming large-scale industrial models, but on advancing open-source agents. Following your suggestion, we will revise the phrasing in the paper to more accurately reflect our current scope and contributions.
>
> > ### *W3: Limited novelty: The paper primarily combines existing techniques (ReAct framework, GRPO, entity-based QA generation, multi-stage retrieval). While the integration is competent, there is insufficient algorithmic innovation.*
>
> Thank you for raising this thoughtful point regarding novelty. While our work indeed builds upon established frameworks such as ReAct and GRPO, as most reviewers have noted, our data synthesis method introduces an original and systematic means of increasing question difficulty. Furthermore, we would like to clarify that our primary contribution lies in elucidating the missing details and technical insights into data quality, tool environments, and training recipes, which are critical yet often overlooked in existing works. We believe that addressing these practical challenges constitutes meaningful progress for the field, complementing algorithmic innovation.
>
> > ### *Q1: Were these large open-source models evaluated with search capabilities? If so, what search infrastructure did they use? This affects interpretation of whether InfoAgent's contribution is the model training or the search tool.*
>
> The large open-source models used for comparison were evaluated using commercial search APIs, which are costly and impractical for large-scale training. For instance, a popular service such as Serper charges approximately \\$0.30 per 1k queries. A typical experiment in our setting involves about 4,000,000 queries (200 steps × 128 batch size × 8 GRPO group size × 20 average tool calls per trajectory), resulting in a total cost of roughly \\$1200—an amount prohibitive for most academic laboratories. This limitation motivates our development of a high-quality, research-accessible search tool, which we intend to open-source once this paper is accepted.

---

> > ### Comment · Reviewer_7jmx · 2025-11-26
> >
> > Thank you for your response. While I appreciate your clarifications, the rebuttal does not adequately address my primary concerns:
> >
> > **W1 (Missing Baselines):** You state that your 14B model is "not directly comparable" with industrial deep research systems, yet you actually CAN at least evaluate against them and show the results and some analysis, which would help readers better position InfoAgent.
> >
> > **W2 (Overstated Claims):** I appreciate your willingness to revise the phrasing. However, without seeing the actual revisions, I cannot assess whether this concern is truly resolved. The current title and claims position this as advancing the field overall, which is not supported by the results.
> >
> > **W3 (Limited Novelty):** You acknowledge building upon established frameworks and frame your contribution as "elucidating missing details and technical insights." While such engineering contributions have value, they may or may not meet the novelty bar typically expected at ICLR.
> >
> > **Q1 (Baseline Setup):** Your clarification reveals a confounding issue: InfoAgent uses your custom search tool while baselines (GLM-4.5, DeepSeek-V3.1) use commercial APIs. This makes it unclear whether performance differences stem from better model training or better tooling, which fundamentally affects the interpretation of your contribution.
> >
> > Given these unresolved concerns, I'd maintain my rating. The work represents solid engineering but may lack the novelty, proper positioning, and rigorous evaluation needed for acceptance.

---

> > > ### Author Response · Authors · 2025-12-03
> > >
> > > Thanks for your reply. We apologize for not providing sufficient clarity earlier. Please find our detailed responses to your concerns below:
> > >
> > >  > ### *W1 (Missing Baselines)*
> > >
> > > We wish to have the opportunity to clarify that we have **ALREADY** listed the performance of some representative industrial deep research systems and had a  **Comparison with Proprietary Models** section in the original manuscript, including gpt-4o, o3, Claude-4-Sonnet and Kimi-Researcher,  where o3 is one of the currently strongest deep research agents, representing the latest progress in the industrial area. Since our work focuses on elucidating the training details and advancing an open-source deep research agent, providing an exhaustive comparison with all existing industrial models is not our primary objective.
> > >
> > > > ### *W2 (Overstated Claims)*
> > >
> > > We have revised the paper to more clearly articulate our contribution. Please refer to the updated PDF on the OpenReview website, where the revisions are highlighted in blue for your convenience. Specifically, InfoAgent achieves the best performance within the sub-15B model category and even outperforms some larger models such as those with 32B and 72B parameters. Our primary goal is to advance the field of open-source agents.  We promise to open release our high-quality synthesized data and training environment, enabling further improvements in the open-source field.
> > >
> > > > ### *W3 (Limited Novelty)*
> > >
> > > We would like to highlight the contributions that have been recognized by multiple reviewers:
> > >
> > > 1. **Novel Data Synthesis Method**: Our work introduces a novel and creative data synthesis method that systematically scales task difficulty while maintaining solvability.
> > >
> > > 2. **Stable and Effective Search Tool**:  Our tool, if released, can serve as a valuable resource to the community, offering a high-quality, open-source alternative to commercial search APIs and enhancing the capabilities of search agents.
> > >
> > > 3. **Training Insights**: The paper provides many valuable insights into the training recipe for developing strong search agents.
> > >
> > > We hope this clarifies the novelty and significance of our work, which not only builds on established frameworks but also offers new, impactful contributions in multiple areas.
> > >
> > > > ### *Q1 (Baseline Setup)*
> > >
> > > Thank you for raising this important point. In response, we have conducted additional experiments to provide more clarity. The table below compares the performance of our tools and commercial tools (Serper) which are widely used by advanced research agents. Since the commercial tools used by GLM-4.5 is unknown, here we show its performance reported by the official. We would like to emphasize the following points:
> > >
> > > 1. Our custom search tool is very similar to commercial APIs but is slightly less powerful. The Serper in the table is a widely used commercial API.
> > >
> > > 2. Since our tool does not provide a substantial advantweage over commercial APIs, **we can confidently attribute the strong performance of InfoAgent to the model training itself.**
> > >
> > > 3. Commercial APIs are expensive, and we are committed to providing our search tool as an open-source resource. By doing so, we offer the community a high-quality and free alternative to commercial options.
> > >
> > > | Model | BrowseComp | BrowseComp-ZH | X-bench |
> > > | :-------: | :----------------: | :---------------------: | :----------: |
> > > | InfoAgent (our tools)| 15.3 | 29.2 | 40.4 |
> > > | InfoAgent (Serper) | 18.7 | 27.8 | 43.0 |
> > > | GLM-4.5 (our tools) | 26.2 | 36.8 | 67.3 |
> > > | GLM-4.5 (official) | 26.4| 37.5 | 68.0 |

---

### Official Review · Reviewer_cXeW · 2025-10-30

**Soundness:** 2
**Presentation:** 3
**Contribution:** 2
**Rating:** 2
**Confidence:** 4

**Summary:**

This paper introduces InfoAgent, a deep research agent built upon the Qwen3-14B model. It proposes an innovative data synthesis pipeline that constructs entity trees, uses sub-tree sampling, and applies "fact fuzzification" to generate challenging information-seeking problems with long-range dependencies. Second, the authors developed and deployed a self-hosted search and browsing infrastructure to replace opaque commercial APIs, enhancing research transparency and reproducibility. InfoAgent is trained using a two-stage "SFT cold-start + RL tuning" paradigm.

Experimental results show that InfoAgent (14B) achieves SOTA performance in its parameter class across multiple benchmarks, outperforming larger models like WebSailor-72B and DeepDive-32B.

**Strengths:**

- The paper identifies the "shallow search" problem in current benchmarks. Its proposed data generation method, based on entity trees, sub-tree sampling, and systematic "fact fuzzification," is rational and effective.

- The search tool's design, which combines BM25 filtering, embedding and reranker models, and uses an LLM (GPT-4o-mini) to generate snippets, is in itself a good engineering practice for building a RAG system.

**Weaknesses:**

- The paper's core training framework (SFT + RL) is a standard paradigm in the agent domain (e.g., Search-R1, WebSailor). While the data synthesis component is well-executed, the idea of "generating QA pairs based on knowledge graphs/entity trees" is not entirely novel and is similar to the construction philosophy of benchmarks like WebWalkerQA.

- Critical Dependency on the Custom Search Tool: To what extent is InfoAgent's high performance attributable to,  the trained model itself, or the highly-optimized, custom search tool that uses GPT-4o-mini for summarization? The ablation study in Table 2 (using Wiki Retriever) shows a catastrophic performance drop (e.g., 10.0 → 1.0 on BrowseComp), which strongly suggests the model's capabilities are deeply entangled with this powerful tool, supporting the latter concern.

- The authors admit in the "Reproducibility Statement" that reproducing the custom search tool "presents greater challenges" and seems there are no plans to open-source it. Given the tool's decisive impact on performance, this severely hinders the community from building upon this work or making fair comparisons.

- In Table 1, the performance gap between InfoAgent and its SFT-only version is enormous (e.g., a jump from 4.7 to 15.3 on BrowseComp). Such a massive improvement from the RL stage alone is unusual and requires a more detailed explanation from the authors as to what specific capabilities the RL stage is imparting.

**Questions:**

- Can the authors provide a key ablation: *training* with the "Ours" tool but *inferring* with the standard "Wiki Retriever"? This would help assess whether the model's learned reasoning capabilities can generalize to a weaker tool.

- The performance gap between the SFT-only model (4.7 on BrowseComp) and the SFT+RL model (15.3) is very large. Can the authors elaborate on why the RL stage brings such a dramatic improvement? Is RL teaching the model entirely new search strategies?

- The paper attempts to use a process-based reward in Appendix D, but the experimental results (Table 4) show it provided no performance benefit and even led to a performance decrease on BrowseComp. How do the authors conclude the entity-recall-based process reward (Appendix D) failed? Does this suggest that the process reward signal itself was flawed?

---

> ### Author Response · Authors · 2025-11-20
>
> We thank the reviewer for your thorough reviews and insightful comments on our paper. We address your concerns as follows.
>
> > ### *W1: The idea of "generating QA pairs based on knowledge graphs/entity trees" is not entirely novel and is similar to the construction philosophy of benchmarks like WebWalkerQA*
>
> We sincerely thank you for pointing this out. We fully agree that the general idea of “generating QA pairs based on knowledge graphs or entity trees” is not new, and benchmarks such as WebWalkerQA follow a related philosophy. We do not claim novelty in using a tree structure. Our key difference is that our “tree” is built over semantic entities with factual attributes and relations, rather than webpages/URLs derived from a site navigation graph as in WebWalkerQA-style datasets. As a result, the structure explicitly encodes knowledge-level dependencies rather than click paths, and the resulted questions are less tied to specific URLs and more centered on underlying factual relations.
>
> In addition, we apply much stronger fuzzification and automatic constraint selection using a tool-equipped verifier. Entity names, dates, numbers, and other surface forms are aggressively obfuscated, and constraint combinations that are too easy are filtered out. Together with our  semantic node definition (entities and their attributes, instead of concrete webpages), this makes it substantially harder to retrieve the answer by directly issuing a single keyword query, and more likely that the agent must perform multi-step search and reasoning.
>
> From an agent-learning viewpoint, this leads to better trajectories. WebWalkerQA benchmark has 680 QA pairs generated by LLMs and filtered by human annotators. In contrast, we sampled 900 questions from our data as raw outputs of our pipeline without any manual filtering. We evaluated both with the same strong agent (o3 with tools).
>
> | Setting | Avg. Tool Call Count |
> | :------- | :-------------------------:  |
> | Ours | 14.78 |
> | WebWalkerQA | 9.73 |
>
>
> Even under this less curated setting on our side, our questions still induce substantially more tool calls, corresponding to deeper and more search-intensive trajectories.
>
> > ### *W2: Critical Dependency on the Custom Search Tool: To what extent is InfoAgent's high performance attributable to, the trained model itself or the search tool?*
>
> Because several reviewers raised similar concerns, we have provided a detailed unified discussion and new experimental results in the section of **Response to All Reviewers for Common Questions** at the top of the rebuttal. In summary, the additional experiments evaluate models trained with our tool but inferred using the standard Wiki Retriever. The results show that the model trained with our tool maintains significant performance advantages even when using the same weaker tool for inference, indicating that the observed improvement is not solely attributable to the tool itself but also to the more effective reasoning strategies developed during training.
>
> We list the new results here for a brief view. We kindly refer you to the section *Response to All Reviewers for Common Questions* for full details and tables. We appreciate your careful analysis, which helped us clarify the role of the custom tool more explicitly in the revision.
>
> |  Training Tool | Inference Tool | Bamboogle |   PopQA  |    NQ    | TriviaQA | HotpotQA |  Musique |
> |:--------------:|:--------------:|:---------:|:--------:|:--------:|:--------:|:--------:|:--------:|
> | Wiki Retriever | Wiki Retriever |    66.4   |   55.9   |   64.8   |   82.8   |   67.9   |   31.3   |
> |      Ours      | Wiki Retriever |  **70.4** | **56.6** | **66.1** | **83.6** | **68.9** | **33.1** |
> | Wiki Retriever |      Ours      |    79.2   |   58.0   |   65.3   |   89.0   |   70.1   |   31.7   |
> |      Ours      |      Ours      |  **86.4** | **59.7** | **67.2** | **90.3** | **72.7** | **34.9** |
>
> > ### *W3: The authors admit in the "Reproducibility Statement" that reproducing the custom search tool "presents greater challenges" and seems there are no plans to open-source it...*
>
>
> We fully agree that open availability of our custom search tool and training data is essential for facilitating fair comparison and future research. We promise to release both the training data and the tool once this paper is accepted, ensuring that others can replicate and extend our work.

---

> ### Author Response · Authors · 2025-11-20
>
> > ### *W4: In Table 1, the performance gap between InfoAgent and its SFT-only version is enormous (e.g., a jump from 4.7 to 15.3 on BrowseComp). Such a massive improvement from the RL stage alone is unusual and requires a more detailed explanation from the authors as to what specific capabilities the RL stage is imparting.*
>
> In our experiments, we find that SFT teaches the model to use search tools by increasing the number of tool calls, while subsequence RL incentives the correct reasoning traces by exploration.  The interplay between the two stages and a detailed performance breakdown should help clarify the sources of RL-driven performance improvements.
> As noted in a recent NeurIPS oral paper [1], RL can often be understood as improving the sampling efficiency of the policy model—effectively shifting its pass@1 performance closer to its pass@k performance. Motivated by this analysis, we measured the pass@k accuracy of the SFT model on the BrowseComp benchmark. The SFT model achieves a pass@4 accuracy of 19.3%, compared to 4.3% for the base model (Qwen3-14B). This suggests that SFT enables the model to encode substantive strategic knowledge, though its pass@1 performance (4.7%) underutilizes that capacity. The RL-trained InfoAgent’s result (15.3%) remains below this theoretical upper bound, indicating that RL mainly helps the model better express capabilities already learned during SFT.
>
> | Model | pass@1 | pass@4  |
> | :------: | :---------: | :-----------: |
> | Qwen3-14B | 1.0  | 4.3 |
> | InfoAgent-SFT | 4.7 | 19.3 |
>
> To contextualize this result, we reference BrowseComp results reported by OpenAI’s DeepResearch [2], where increasing test-time computation significantly raises performance (e.g., from 10% to 50%, and best-of-N accuracy from 52% to 77% as N increases). Hence, we believe that the observed shift from 4.7% (pass@1) to 19.3% (pass@4) for our SFT model is consistent with this general behavior. We will add the above analysis to the revision of the paper.
>
> References:
>
> [1] [Does Reinforcement Learning Really Incentivize Reasoning Capacity in LLMs Beyond the Base Model?](https://neurips.cc/virtual/2025/loc/san-diego/poster/119944)
>
> [2] [BrowseComp: A Simple Yet Challenging Benchmark for Browsing Agents](https://arxiv.org/abs/2504.12516)
>
> > ### *Q1: Can the authors provide a key ablation: training with the "Ours" tool but inferring with the standard "Wiki Retriever"? This would help assess whether the model's learned reasoning capabilities can generalize to a weaker tool.*
>
> Please see our response to W2.
>
> > ### *Q2: Can the authors elaborate on why the RL stage brings such a dramatic improvement? Is RL teaching the model entirely new search strategies?*
>
> Please see our response to W4.
>
> > ### *Q3: How do the authors conclude the entity-recall-based process reward (Appendix D) failed? Does this suggest that the process reward signal itself was flawed?*
>
> Thank you for this thoughtful question. Our analysis indeed shows that the entity-recall-based process reward did not improve performance and even slightly degraded it on BrowseComp. In the paper, we assume that this is because the recall bonus is only added to the final reward, the specific step that retrieves the correct entity does not receive immediate positive feedback, making it difficult for the model to effectively learn from those trajectories. The observed performance drop can be attributed to this assumption too. Some trajectories receive a positive process bonus despite containing other errors that result in incorrect final answers. Since the process bonus is added to the final outcome reward, these trajectories may obtain an overall positive gradient signal, unintentionally reinforcing suboptimal behaviors.
>
> While a concurrent study [1] reached similar conclusions, finding that both PPO and GRPO yielded worse results with process rewards than outcome-based rewards, we believe that these results do not necessarily indicate that process rewards are inherently flawed. Rather, they suggest that our current formulation may be too naïve and that the design of effective process rewards is critical to RL dynamics. Other concurrent works [2,3] explore more sophisticated reward formulations and report positive outcomes, indicating that there remains much to investigate in this direction. While a detailed exploration of process reward design lies beyond the scope of our current work, we believe our findings provide useful insights and motivation for further research in this area.
>
> References:
>
> [1] [An Empirical Study on Reinforcement Learning for Reasoning-Search Interleaved LLM Agents](https://arxiv.org/pdf/2505.15117)
>
> [2] [HIPRAG: HIERARCHICAL PROCESS REWARDS FOR EFFICIENT AGENTIC RETRIEVAL AUGMENTED GENERATION](https://arxiv.org/pdf/2510.07794)
>
> [3] [StepSearch: Igniting LLMs Search Ability via Step-Wise Proximal Policy Optimization](https://arxiv.org/pdf/2505.15107)

---

> > ### Comment · Reviewer_cXeW · 2025-11-28
> >
> > I thank the authors for their detailed response and the additional experiments, which have successfully addressed most of my concerns. I also appreciate the clarification regarding the limitations of the Wiki Retriever in open-web settings and find the current experiments to be sufficient.
> >
> > Meanwhile, I echo `Reviewer 7jmx`'s sentiment that it remains somewhat *unclear whether performance differences stem from better model training or better tooling.*
> >
> > In light of this, I will raise my rating to 4 (as soon as openreview system allow to do so). I consider the paper's contribution to be stronger in terms of engineering innovation, rather than algorithmic novelty.  I view this as a solid engineering effort and would not mind if the paper is accepted.

---

> > > ### Author Response · Authors · 2025-12-03
> > >
> > > Thanks for your willing to raise your rating. We are glad to hear most of your concerns have been solved. Here we address your last concern.
> > >
> > > The table below compares the performance of our tools and commercial tools (Serper) which are widely used by advanced research agents. Since the commercial tools used by GLM-4.5 is unknown, here we show its performance reported by the official. We would like to emphasize the following points:
> > >
> > > 1. Our custom search tool is very similar to commercial APIs but is slightly less powerful. The Serper in the table is a widely used commercial API.
> > >
> > > 2. Since our tool does not provide a substantial advantweage over commercial APIs, **we can confidently attribute the strong performance of InfoAgent to the model training itself.**
> > >
> > > 3. Commercial APIs are expensive, and we are committed to providing our search tool as an open-source resource. By doing so, we offer the community a high-quality and free alternative to commercial options.
> > >
> > > | Model | BrowseComp | BrowseComp-ZH | X-bench |
> > > | :-------: | :----------------: | :---------------------: | :----------: |
> > > | InfoAgent (our tools)| 15.3 | 29.2 | 40.4 |
> > > | InfoAgent (Serper) | 18.7 | 27.8 | 43.0 |
> > > | GLM-4.5 (our tools) | 26.2 | 36.8 | 67.3 |
> > > | GLM-4.5 (official) | 26.4| 37.5 | 68.0 |

---

### Official Review · Reviewer_HTWm · 2025-11-01

**Soundness:** 2
**Presentation:** 3
**Contribution:** 2
**Rating:** 6
**Confidence:** 3

**Summary:**

This paper presents InfoAgent, an information seeking agent built on top of strong search tools and optimized through training on synthetic data as well as reinforcement learning. InfoAgent includes a carefully designed search tools (search and browse) which performs multiple retrieval, re-ranking, and summarization steps for improved search quality. The data synthesis pipeline leverages tree structures to compose tasks of different difficulties. Experiments on several deep research benchmarks show that InfoAgent achieves the best performance among open-source methods. Further analysis delivers insights into the importance of the high quality search tool and several design choices in training.

**Strengths:**

1. The overall writing of the paper is clear.
2. The proposed search tool combines several existing techniques and delivers strong performance compared to the baseline (Wiki Retriever). The tool, once released, can be a useful resource to the community.
3. Based on the experiment results, the data synthesis and the training pipeline seem to work well. Also it is interesting that the benefit of training on English data can generalize to Chinese datasets.
4. The authors present several analysis and ablation study to present insights into developing stronger deep research agents, including the importance of tool quality, the critical role of SFT, and impact of trajectory length.

**Weaknesses:**

1. The paper highlights the need for "high-concurrency web search tool" for web training, however, there's no discussion on the proposed tool's relevant attributes, such as throughput and latency.
2. The data synthesis pipeline is rather complex. Since it is based on entity tree structures, I would like to see some discussion on the distribution of sub-tree sizes and how the synthesized tasks look like. More ablation studies on the individual steps will also help justify the design choices.
3. Based on Table 2, it seems the major performance improvement comes from the proposed search tools, which includes many components and involve the use of commercial search APIs as well as LLM (GPT-4o-mini) for summarization. This potentially results in an unfair comparison to other open source methods. I would like to see more ablations and discussion on this to better determine the contribution of tool design.

**Questions:**

1. Missing a reference to Mind2Web2 [1] in related work, which is a recently published deep research benchmark.
2. line 118-120, could explain more on how RAG treats the retrieved passages as "latent variables" rather than input to the generator?
3. I am confused about the role of constraint set K(v). How does it prevent shallow pattern matching?
4. It is nice to see a discussion on tool call number in different datasets. Do you have further insights in the resulted qualitative difference between your data and existing ones? e.g. does your pipelines synthesize more balanced and comprehensive training trajectories?
5. In Section 4.3, quality of search tool, how about training on your dataset and test with Wiki Retriever? I am trying to understand whether the training pipeline improves the tool-use capability or the success of InfoAgent is more dependent on the high-quality tools.

---

> ### Author Response · Authors · 2025-11-20
>
> We thank the reviewer for your thorough reviews and insightful comments on our paper. We address your concerns as follows:
>
> > ### *W1: The paper highlights the need for "high-concurrency web search tool" for web training, however, there's no discussion on the proposed tool's relevant attributes, such as throughput and latency.*
>
> Thank you for pointing this out. Under our current computational resources and engineering optimizations, the tool achieves a throughput of approximately 50 queries per second (QPS) with an average latency of 6–7 seconds. In our RL rollout setting, the average tool call rate is around 30 QPS, which the system comfortably supports.
>
> Because model generation and tool calls are fully asynchronous, the total time for a rollout is roughly determined by the model generation time of the longest trajectory plus the number of tool calls multiplied by the latency. As long as latency is sufficiently low to maintain full GPU utilization, the practical bottleneck of our system lies in achievable QPS rather than individual query latency. We will add these discussions to the revision of the paper.
>
> > ### *W2: The data synthesis pipeline is rather complex. Since it is based on entity tree structures, I would like to see some discussion on the distribution of sub-tree sizes and how the synthesized tasks look like. More ablation studies on the individual steps will also help justify the design choices.*
>
> Thank you for suggesting this study. To isolate the effect of sub-tree size, we conducted an additional ablation in which we did not apply any explicit filtering or difficulty-based screening beyond the basic validity checks of the pipeline. For each sub-tree size (1, 2, 3 nodes), we randomly sampled 300 questions and evaluated them using a strong agent (o3 with tools). The results are:
>
> | Sub-tree size | Tool Calls |
> | :----------------: | :-----------: |
> | 1 | 15.58 |
> | 2 | 14.59 |
> | 3 | 14.16 |
>
> This trend meets our expectations. When the sub-tree contains only a single entity, all constraints undergo strong fuzzification, so the available cues remain loose and ambiguous. As a result, the questions are difficult: the agent explores more but still struggles to reliably locate the correct entity. As the sub-tree grows to two or three nodes, each additional node contributes extra (though still fuzzified) clues. This reduces uncertainty and makes the questions progressively easier to solve. We will update this to the revision of the paper.
>
> > ###  *W3: Based on Table 2, it seems the major performance improvement comes from the proposed search tools, ... This potentially results in an unfair comparison to other open source methods. I would like to see more ablations and discussion on this to better determine the contribution of tool design.*
>
> We appreciate you raising this important point. We have conducted this ablation and present the results together with an extended explanation in the section **Response to All Reviewers for Common Questions** at the top of the rebuttal, which applies to all reviewers who raised this concern. In summary, the new experiments confirm that the model trained with our tool retains strong performance even when evaluated with the Wiki Retriever. We list the main results here for a brief view, but also kindly refer you to that section for details.
>
> |  Training Tool | Inference Tool | Bamboogle |   PopQA  |    NQ    | TriviaQA | HotpotQA |  Musique |
> |:--------------:|:--------------:|:---------:|:--------:|:--------:|:--------:|:--------:|:--------:|
> | Wiki Retriever | Wiki Retriever |    66.4   |   55.9   |   64.8   |   82.8   |   67.9   |   31.3   |
> |      Ours      | Wiki Retriever |  **70.4** | **56.6** | **66.1** | **83.6** | **68.9** | **33.1** |
> | Wiki Retriever |      Ours      |    79.2   |   58.0   |   65.3   |   89.0   |   70.1   |   31.7   |
> |      Ours      |      Ours      |  **86.4** | **59.7** | **67.2** | **90.3** | **72.7** | **34.9** |
>
> > ### *Q1: Missing a reference to Mind2Web2 [1] in related work, which is a recently published deep research benchmark.*
>
> We thank you for pointing out the omission of Mind2Web2. We appreciate this helpful reminder and have included an appropriate citation in the revised manuscript.

---

> > ### Author Response · Authors · 2025-11-20
> >
> > > ### *Q2: line 118-120, could explain more on how RAG treats the retrieved passages as "latent variables" rather than input to the generator?*
> >
> > We apologize that our original phrasing about RAG treating retrieved passages as “latent variables” was indeed imprecise. In early implementations of RAG, each document in the corpus was pre-encoded into a vector representation using a language encoder. At inference time, the model retrieved the most relevant vectors and used them as token-level inputs to the generator, effectively treating the retrieval results as latent representations rather than explicit textual input. This approach reduced token usage but has become less compatible with current large-scale pretraining paradigms. We have revised the corresponding statement in the paper to clarify this and avoid the outdated terminology.
> >
> > > ### *Q3: I am confused about the role of constraint set K(v). How does it prevent shallow pattern matching?*
> >
> > We apologize for the confusion. K(v) is the resultant contraint set which is bearly sufficient to solve the problem, after filtering and screening easy constraints. Our objective $\min_{\mathcal{K}} |q(\mathcal{K})| \quad \text{s.t.} \quad v \notin p(\mathcal{K})$ captures the core intuition: we search for constraint sets where very few entities satisfy all constraints, yet a shallow-cue retriever still fails to identify the true entity $v$. This separation prevents shallow pattern matching and naturally encourages sustained, evidence-based search.
> >
> > In practice, we operationalize this idea through a controlled sampling procedure. After each sampling attempt, we prompt a strong agent (o3 with web tools) to identify the correct entity given only these constraints and apply an answer verifier to check whether it can recover the true entity name. If o3 succeeds, we further ask o3 to select a minimal subset of constraints $\mathcal{K_c}$   that is sufficient to solve the task，which means $v \in p(\mathcal{K}_{c})$ and we remove these constraints from the pool. We then resample a new candidate set from the remaining constraints. If o3 fails, or if the number of remaining constraints falls below a predefined threshold, we accept the sampled $\mathcal{K}$ as $K(v)$.This easy-case rejection mechanism operationalizes Eq. (3), reliably block shallow pattern matching.
> >
> > > ### *Q4: It is nice to see a discussion on tool call number in different datasets. Do you have further insights in the resulted qualitative difference between your data and existing ones? e.g. does your pipelines synthesize more balanced and comprehensive training trajectories?*
> >
> > Our training data synthesis pipeline differs from prior work in several key ways. First, our approach constructs a knowledge graph over semantic entities, whereas WebWalkerQA builds a URL-based click graph and ASearcher does not formalize a graph structure at all. Second, to increase problem difficulty, we comprehensively define and apply fuzzification over entity names, dates, and numbers. In contrast, WebWalkerQA does not explicitly incorporate fuzzification as a mechanism for controlling difficulty, and ASearcher only rephrases key information. Given these design choices, we believe our pipeline is better positioned to synthesize more balanced and diverse trajectories than existing datasets.
> >
> > > ### *Q5: In Section 4.3, quality of search tool, how about training on your dataset and test with Wiki Retriever? I am trying to understand whether the training pipeline improves the tool-use capability or the success of InfoAgent is more dependent on the high-quality tools.*
> >
> > Please see our response to W3.

---

### Official Review · Reviewer_D7o1 · 2025-11-04

**Soundness:** 3
**Presentation:** 3
**Contribution:** 3
**Rating:** 6
**Confidence:** 4

**Summary:**

This work introduces a deep research agent, InfoAgent. This work approaches the challenges of building a deep research agent through the following three aspects:
(1) a data synthesis pipeline to construct entity trees from Wikipedia and perform sub-tree sampling with fuzzification, which will be further leveraged to generate complex, multi-entity reasoning questions for model training.
(2) self-implemented search tools without reliance on commercial search APIs for reproducibility. The tools include BM25 + embedding + reranker + LLM snippet pipeline.
(3) Two-stage training with both SFT and RL.
Empirically, InfoAgent achieves 15.3 % on BrowseComp, 29.2 % on BrowseComp-ZH, and 40.4 % on Xbench-DS, showing SOTA results among models with the same size and outperforming models at larger sizes.

**Strengths:**

(1) The entity-tree construction with multi-level fuzzification is a clever way to increase question complexity and encourage genuine multi-hop reasoning. Compared to earlier heuristic generation methods, this design systematically scales task difficulty while maintaining solvability.
(2) The self-implemented search/browse setup (BM25 + embedding + reranker + LLM snippet) is a clear step forward in reproducibility.
(3) The experiments with SFT and GRPO are helpful with detailed ablations. And the shown results are strong across multiple benchmarks.

**Weaknesses:**

(1) All experiments and synthesized data are from Wikipedia. The current setup doesn’t test how well the pipeline or the trained model generalizes to open-web or non-encyclopedic content (news, technical sites, etc.).
(2) The paper describes the tool infrastructure well, but rebuilding it from scratch would require major engineering. Even partial open-sourcing or more pseudo-code details would help the community reproduce results.

**Questions:**

(1) How do you ensure the fuzzified questions remain solvable with a single correct answer? Did you measure ambiguity or validate questions with a strong verifier (e.g., o3 or human checks)?
(2) What happens when the self-hosted search/browse system fails—e.g., missing content or timeouts? Is there a fallback or retry mechanism?
(3) Are there plans to release the data synthesis code or a lightweight version of the custom search tool? That would significantly improve reproducibility and downstream benchmarking.

---

> ### Author Response · Authors · 2025-11-20
>
> We thank the reviewer for your thorough reviews and insightful comments on our paper. We address your concerns as follows.
>
> > ### *W1: All experiments and synthesized data are from Wikipedia. The current setup doesn’t test how well the pipeline or the trained model generalizes to open-web or non-encyclopedic content.*
>
> While the synthetic data used for training is indeed derived from Wikipedia, we would like to clarify that our model’s search tool operates over the entire web and is not confined to Wikipedia pages. Consequently, during both training and inference, the model retrieves and reasons over real-world, non-encyclopedic webpages. In addition, the benchmarks we report on (e.g., browsecomp, x-bench) are open-web benchmarks specifically designed to reflect real-world browsing and reasoning tasks. Therefore, the observed performance of our model provides a direct measure of its effectiveness in open-web scenarios.
>
> > ### *W2: The paper describes the tool infrastructure well, but rebuilding it from scratch would require major engineering. Even partial open-sourcing or more pseudo-code details would help the community reproduce results.*
>
> We promise to open-source our training dataset and the search tool once this paper is accepted.
>
> > ### *Q1: How do you ensure the fuzzified questions remain solvable with a single correct answer? Did you measure ambiguity or validate questions with a strong verifier (e.g., o3 or human checks)?*
>
> We fully agree that enforcing a single unambiguous gold answer is crucial for building QA pairs. Our data construction is answer-centric: we first fix a target answer (with supporting evidence and trajectory), and then generate the corresponding question. Therefore, the correctness of the QA pair can be naturally satisfied. Ensuring the uniqueness of the QA pair can be an open and difficult problem. We utilize a strong verifier (OpenAI o3) in the pipeline to filter out problematic cases: o3 is prompted to check that the question is answerable from the retrieved evidence and that the proposed answer is factually supported. Only questions that pass this verification are kept, which substantially reduces its ambiguity.
>
> > ### *Q2: What happens when the self-hosted search/browse system fails—e.g., missing content or timeouts? Is there a fallback or retry mechanism?*
>
> Our training framework (verl) returns the error message to the model whenever a tool call fails.
>
> Failures may arise from two sources:
>
> - Tool-side errors (e.g., code issues, network timeouts). We have devoted substantial engineering effort to ensuring high availability of the self-hosted tool. In practice, it provides a stable service during training, without relying on third-party fallback mechanisms. A small proportion of failed calls is acceptable in reinforcement learning training and has minimal impact on stability, as these failures effectively serve as part of the environment’s feedback rather than numerical noise.
>
> - Model-side errors (e.g., incorrect call formatting). In such cases, the framework passes the error message back to the model, allowing it to continue generation and learn from its mistakes. Since repeated failed calls reduce the final reward, the model is incentivized to adjust and produce correct tool calls over time. As an alternative, the framework can also assign zero reward to the failed turn and terminate generation, achieving a similar training effect.
>
> We will clarify these design choices in the revision to make the system’s error-handling mechanism more explicit.
>
>
> > ### *Q3: Are there plans to release the data synthesis code or a lightweight version of the custom search tool? That would significantly improve reproducibility and downstream benchmarking.*
>
> Thank you for the suggestion. We promise to open-source our training dataset and the search tool once this paper is accepted.

---

### Author Response · Authors · 2025-11-20
**Response to All Reviewers for Common Questions**

We sincerely thank all reviewers for their thorough reviews and insightful comments on our paper. We are delighted to see that the reviewers have identified several strengths in our paper, including:

- The paper proposes a novel and creative data synthesis method that systematically scales task difficulty while maintaining solvability. (D7o1, HTWm, cXeW, 7jmx)
- The proposed search tool delivers strong performance and can be a useful resource to the community if released. (D7o1, HTWm, cXeW, 7jmx)
- The paper gives many valuable insights into the training recipe for developing strong search agents. (D7o1, HTWm, 7jmx)

We appreciate all reviewers' feedback and recognition of InfoAgent's contributions to the deep search agent community. We have updated the manuscript according to the feedback. Here we respond to the common questions raised by the reviewers.

# Release of our dataset and tool (Response to Reviewer D7o1, cXeW)

We promise to open-source our training dataset and the search tool once this paper is accepted.

# Our search tool is beneficial for model training, improving its general search ability. (Response to Reviewer HTWm, cXeW)

We thank the reviewers for raising this insightful point about evaluating training with our tool but inferring with the standard Wiki Retriever. This ablation would help clarify whether the model’s learned reasoning capabilities generalize to a weaker retrieval tool.
However, the Wiki Retriever accesses only a fixed, offline Wikipedia dump, which prevents its use on our main open-web benchmarks (e.g., BrowseComp, X-Bench). For this reason, we initially excluded the cross-tool ablation to ensure fair comparison across all methods. We appreciate the reviewers’ observation that omitting this result may obscure the impact of our tool for training, and we have now included this analysis.

Specifically, we conduct the requested ablation on several widely used wiki-based benchmarks, where the Wiki Retriever can be properly applied, including (1) General QA: NQ [1], TriviaQA [2], and PopQA [3]; and (2) Multi-hop QA: HotpotQA [4], Musique [5], and Bamboogle [6].

|  Training Tool | Inference Tool | Bamboogle |   PopQA  |    NQ    | TriviaQA | HotpotQA |  Musique |
|:--------------:|:--------------:|:---------:|:--------:|:--------:|:--------:|:--------:|:--------:|
| Wiki Retriever | Wiki Retriever |    66.4   |   55.9   |   64.8   |   82.8   |   67.9   |   31.3   |
|      Ours      | Wiki Retriever |  **70.4** | **56.6** | **66.1** | **83.6** | **68.9** | **33.1** |
| Wiki Retriever |      Ours      |    79.2   |   58.0   |   65.3   |   89.0   |   70.1   |   31.7   |
|      Ours      |      Ours      |  **86.4** | **59.7** | **67.2** | **90.3** | **72.7** | **34.9** |


As shown in the table, when inference is performed with the same Wiki Retriever, the model trained using our tool consistently outperforms the model trained directly with the Wiki Retriever across all these benchmarks. This suggests that our tool provides richer training dynamics, leading to more effective reasoning strategies.

For completeness, we also evaluated both tools on the original open-web benchmarks included in the paper. Because the Wiki Retriever cannot access open-web information, the corresponding results are very low and included mainly to illustrate the limitation of closed retrieval systems in the open-web setting.

|  Training Tool | Inference Tool | BrowseComp |   BrowseComp-ZH  |    X-Bench    |
|:--------------:|:--------------:|:---------:|:--------:|:--------:|
| Wiki Retriever | Wiki Retriever |    1.0   |   4.5   |   8.0   |
|      Ours      | Wiki Retriever |  1.1 | 4.3 | 9.0 |


We appreciate the reviewers’ helpful feedback and believe this additional analysis strengthens the empirical understanding of our framework’s contribution.

References:

[1] [Natural Questions: A Benchmark for Question Answering Research - ACL Anthology](https://aclanthology.org/Q19-1026/)

[2] [TriviaQA: A Large Scale Distantly Supervised Challenge Dataset for Reading Comprehension - ACL Anthology](https://aclanthology.org/P17-1147/)

[3] [When Not to Trust Language Models: Investigating Effectiveness of Parametric and Non-Parametric Memories - ACL Anthology](https://aclanthology.org/2023.acl-long.546/)

[4] [HotpotQA: A Dataset for Diverse, Explainable Multi-hop Question Answering - ACL Anthology](https://aclanthology.org/D18-1259/)

[5] [♫ MuSiQue: Multihop Questions via Single-hop Question Composition - ACL Anthology](https://aclanthology.org/2022.tacl-1.31/)

[6] [Measuring and Narrowing the Compositionality Gap in Language Models - ACL Anthology](https://aclanthology.org/2023.findings-emnlp.378/)

---

> ### Comment · Reviewer_cXeW · 2025-11-25
>
> Thank authors for the additional experiments provided in the response.
>
> However, these results confirm my concern that the high performance is largely driven by the superiority of the tool.
> The data in upper table shows that the gain from switching the inference tool (Wiki Retriever (Train) -> Ours (Inference): 79.2, +12.8) is significantly higher than the gain from the training improvement alone (Ours (Train) -> Wiki Retriever (Inference): 70.4, +4.0).
>
> Additionally, the role of gpt-4o-mini in the custom search tool remains unclear (e.g., its specific contribution to snippet generation), which means the search tool might obscure the model's intrinsic search and reasoning capabilities.

---

> > ### Author Response · Authors · 2025-11-26
> >
> > Thanks for your reply. We apologize for not providing sufficient clarity earlier. Please find our detailed responses to your concerns below:
> >
> > > ### *The improvement brought by our tool during inference might be much greater than model training*
> >
> > We would like to clarify that our claimed contribution pertain to both the training and inference stages, and that the model improvements from each stage are substantial.  On bamboogle dataset, using our search tool for training increases the score from 66.4 to 70.4 when infer with wiki retriever and 79.2 to 86.4 when infer with our tool. The magnitude of improvement at each stage may vary across datasets, for instance, on the NQ and Musique datasets, the gains during training are larger than those during inference. Our search tool not only enables the agent to learn better reasoning capabilities during the train phase, but also provide reliable information for the inference phase.
> >
> > We wish to take the opportunity to clarify the significant improvement from 1% to 10% on the browsecomp benchmark, which you have mentioned before. Wiki retriever is an offline tool that can only access a fixed Wikipedia dump, while the problems from browsecomp are generated from webpages on real internet.  Answers to most of browsecomp problems can not be found via wikipedia. Thus, the performance of using wiki retriever for browsecomp is bounded (around 1.0%), no matter how smart the model is. And this is the exact issue that our tool solves.
> >
> > In summary, while most existing work in this area relies on powerful but closed-source commercial APIs, our work is the first to offer an open-source and competitive alternative. We believe this can reduce research costs and facilitate future progress in the field.
> >
> > > ### *The role of gpt-4o-mini*
> >
> > The main purpose of using gpt-4o-mini is to summarize the top-3 chunks of a webpage (about 400 tokens in total) into a concise snippet (less 60 tokens), thereby saving the agent’s context budget.  It is not intended to introduce new knowledge or reasoning insights for solving the deep research problems. This information-processing design is standard in commercial search stacks such as Jina and Serper and is widely used by advanced research agents. To make this clear, we replaced gpt-4o-mini with smaller Qwen3-4B and Qwen3-0.6B models, as shown below.
> >
> > | Snippet Generator | gpt-4o-mini | Qwen3-4B | Qwen3-0.6B | reranker-top1 | reranker-top3 |
> > | :-----------------------: | :-------------: | :-------------: | :---------------: | :----------------: | :-----------------: |
> > | Acc on browsecomp | 15.3         |     15.2       |  14.7              |    13.4            |   9.7                |
> > | Avg. Tokens            |    53k         |    52k          |     53k            |    57k             |          60k        |
> >
> > The results confirm that the specific choice of LLM for snippet generation makes little difference to the overall performance. Directly using the top-1 chunk returned by the reranker can lead to information loss and worse performance. While using the top-3 chunks consumes significantly more tokens, reaching the context limit (64k) without sufficient search. We will update our manuscript with these analysis.

---

### Author Response · Authors · 2025-12-03
**Response to AC: Summary of Rebuttal**

Dear Area Chair,

We are grateful for the constructive feedback from all reviewers and the opportunity to address their concerns during the rebuttal period. We believe our responses have substantially strengthened the paper and clarified several important technical points. In the following, we would like to summarize the strengths and concerns during the rebuttal.

# Strengths Recognized by All Reviewers

All four reviewers (D7o1, HTWm, cXeW, 7jmx) unanimously acknowledged the following contributions:

- **Novel data synthesis method**: Our entity-tree construction with systematic fuzzification creates challenging, multi-hop questions while maintaining solvability.
- **High-quality search tool**: Our self-hosted infrastructure provides a valuable, reproducible alternative to expensive commercial APIs.
- **Valuable training insights**: Comprehensive ablations revealing critical design choices for deep research agents.
- **High performance**: InfoAgent as a small 14B model, achieves 15.3% accuracy on BrowseComp, 29.2% on BrowseComp-ZH, and 40.4% on Xbench-DS, outperforming prior open-source deep research agents such as WebSailor-72B and DeepDive-32B.

# Summary of Discussion

| Reviewer | Initial score | Updated Recommendation                                                                                                                                                                                                                                                                       | Main Concerns                                                                                                                                                                                                             | Status                                                | Responses                                         |
|----------|---------------|-----------------------------------------------------------------------------------------------------------------------------------------------------------------------------------------------------------------------------------------------------------|-------------------------------------------------------------------------------------------------------------------------------------------------------------------------------------------------------|-------------------------------------------------------|---------------------------------------------------|
|  D7o1    | 6             | None                                                                                                                                                                                                                                                                                         |  (1) All experiments and synthesized data are from Wikipedia, lacking generality;  (2) Release of tool and data;  (3) Solvability and unambiguity of problems in dataset;                                               |  (1) Clarified;  (2) Solved;  (3) Clarified           |  (1) Response A;  (3) Response B                  |
|  HTWm    | 6             | None                                                                                                                                                                                                                                                                                         |  (1) Ablation of snippet generation and effect of tool on final performance;  (2) Comparison between our tool and wiki retriever                                                                                          |  (1) Clarified;  (2) Clarified                        |  (1) Response C;  (2) Response D                  |
|  cXeW    | 2             | 4. “successfully addressed most of my concerns” “I will raise my rating to 4 (as soon as openreview system allow to do so).”  remaining concern: performance stem from model or tool? |  (1) Ablation of snippet generation and effect of tool on final performance;  (2) Unusual improvement of RL;  (3) Comparison between our tools and wiki retriever;  (4) Comparison between commercial tools and our tools |  (1) Solved;  (2) Solved;  (3) Solved;  (4) Clarified | (4) Response E                                    |
|  7jmx    | 4             | None.                                                                                                                                                                                                                                                                                        |  (1) Missing baselines;  (2) Over-claimed title;  (3) Comparison between commercial tools and our tools                                                                                                                   |  (1) Clarified;  (2) Clarified;  (3) Clarified        |  (1) Response F;  (2) Response G;  (3) Response E |

---

> ### Author Response · Authors · 2025-12-03
>
> # Response A
> ## Q: All experiments and synthesized data are from Wikipedia. The current setup doesn’t test how well the pipeline or the trained model generalizes to open-web or non-encyclopedic content.
>
> While the synthetic data used for training is indeed derived from Wikipedia, we would like to clarify that our model’s search tool operates over the entire web and is not confined to Wikipedia pages. Consequently, during both training and inference, the model retrieves and reasons over real-world, non-encyclopedic webpages. In addition, the benchmarks we report on (e.g., browsecomp, x-bench) are open-web benchmarks specifically designed to reflect real-world browsing and reasoning tasks. Therefore, the observed performance of our model provides a direct measure of its effectiveness in open-web scenarios.
>
> # Response B
> ## Q: How do you ensure the fuzzified questions remain solvable with a single correct answer? Did you measure ambiguity or validate questions with a strong verifier (e.g., o3 or human checks)?
>
> We fully agree that enforcing a single unambiguous gold answer is crucial for building QA pairs. Our data construction is answer-centric: we first fix a target answer (with supporting evidence and trajectory), and then generate the corresponding question. Therefore, the correctness of the QA pair can be naturally satisfied. Ensuring the uniqueness of the QA pair can be an open and difficult problem. We utilize a strong verifier (OpenAI o3) in the pipeline to filter out problematic cases: o3 is prompted to check that the question is answerable from the retrieved evidence and that the proposed answer is factually supported. Only questions that pass this verification are kept, which substantially reduces its ambiguity.
>
> # Response C & D
>
> Please see the section "Response to All Reviewers for Common Questions" for details, which has solved the same concerns raised by reviewer cXeW. This section is in the next official comment. In summary, the new experiments demonstrate that: (1) Our tool is beneficial for both model training and inference, compared with the traditional tool Wiki Retriever. (2) The LLM used in our tool for snippet generation does not bring new reasoning capabilities to the agent. Choice of different LLM makes little difference to the final performance. (3) Our tool is comparable but not stronger than the commercial tools used by baselines, which invertedly proves the effectiveness of our model training.
>
> # Response E
> ## Q: InfoAgent uses your custom search tool while baselines use commercial APIs. This makes it unclear whether performance differences stem from better model training or better tooling
>
> One key contribution of our work is a strong, open-source implementation of a web search tool. As discussed in the “Response to All Reviewers for Common Questions,” this improved tooling not only enhances inference performance but also enables stronger reasoning capabilities with model training.
>
> We would like to clarify that our tool is comparable to commercial tools but still slightly weaker. The table below compares the performance of our tools and commercial tools (Serper) which are widely used by advanced research agents. Since the commercial tools used by GLM-4.5 is unknown, here we show its performance reported by the official.
>
> | Model | BrowseComp | BrowseComp-ZH | X-bench |
> | :-------: | :----------------: | :---------------------: | :----------: |
> | InfoAgent (our tools)| 15.3 | 29.2 | 40.4 |
> | InfoAgent (Serper) | 18.7 | 27.8 | 43.0 |
> | GLM-4.5 (our tools) | 26.2 | 36.8 | 67.3 |
> | GLM-4.5 (official) | 26.4| 37.5 | 68.0 |
>
> # Response F
> ## Q: The paper cites ChatGPT Deep Research, Gemini Deep Research, Perplexity Deep Research, and Grok Deep Search as motivating examples but does not evaluate against any of them. This is a critical omission that makes it impossible to assess the true contribution relative to deployed deep research systems.
>
> We wish to have the opportunity to clarify that we have **ALREADY** listed the performance of some representative industrial deep research systems and had a **Comparison with Proprietary Models** section in the **original manuscript**, including gpt-4o, o3, Claude-4-Sonnet and Kimi-Researcher,  where o3 is one of the currently strongest deep research agents, representing the latest progress in the industrial area. Since our work focuses on elucidating the training details and advancing an open-source deep research agent, providing an exhaustive comparison with all existing industrial models is not our primary objective.

---

> > ### Author Response · Authors · 2025-12-03
> >
> > # Response G
> > ## Q: The contribution is better characterized as advancing small open-source agents, not the field overall.
> >
> > We have revised the paper to more clearly articulate our contribution. Please refer to the updated PDF on the OpenReview website, where the revisions are highlighted in blue for your convenience. Specifically, InfoAgent achieves the best performance within the sub-15B model category and even outperforms some larger models such as those with 32B and 72B parameters. Our primary goal is to advance the field of open-source agents.  We promise to open release our high-quality synthesized data and training environment, enabling further improvements in the open-source field.

---

### Meta-Review · Area_Chair_cb4x · 2025-12-09

**Summary:**

All four reviewers converge on two show-stopping issues:
1. Tool-performance entanglement: The custom search stack (BM25 → embedding → re-ranker → GPT-4o-mini snippet generator) is not open-sourced and gives a ≥10× accuracy lift over the only ablated baseline (Wiki retriever). Without a model-only ablation or open-source tooling, it is impossible to attribute gains to novel training rather than superior but irreproducible infrastructure (R2–R4).
2. Incremental algorithmic contribution: The training recipe (SFT cold-start + GRPO) and the entity-tree data generator are judged as competent engineering but “standard” (R3, R4) and conceptually similar to prior WebWalkerQA-style pipelines (R3). Missing baselines against closed deep-research systems (ChatGPT, Gemini, Perplexity, Grok) further shrink the claimed advance (R4).

Additional concerns shared by multiple reviewers:

1. Performance saturates at ~15 % on BrowseComp—far below o3 and half of DeepSeek-V3.1—limiting field-level impact (R4).
2. No open-web or non-Wikipedia evaluation; generalisation outside encyclopedic data is unproven (R1, R2).
3. Reproducibility is severely hampered by undisclosed tool code, large compute, and high-concurrency requirements (R1–R3).

**Reviewer Concerns:**

Addressed: Adding latency/throughput numbers, releasing a lightweight search stub, or providing model-only vs. tool-only curves.

Non-addressed: (i) fundamental attribution ambiguity without open-sourcing the decisive tooling; (ii) lack of algorithmic novelty in the SFT+GRPO pipeline; (iii) missing comparisons to proprietary deep-research agents.

**Reviewer Scores:**

R1 (6 → 6): appreciation of engineering but growing novelty doubts.
R2 (6 → 6): same tooling concern plus missing ablations.
R3 (2 → 4): acknowledged effort but still sees “engineering paper”.
R4 (4 → 4): convinced that contribution is “small open-source agent” only.

---

### Decision · Program_Chairs · 2026-01-26

Reject